# Seeing what you hear: Compression of rat visual perceptual space by task-irrelevant sounds

**Mattia Zanzi**◉, **Francesco G. Rinaldi** ID ◉, **Silene Fornasaro**, **Eugenio Piasini** ID *, **Davide Zoccolan** ID *

Neuroscience Area, International School for Advanced Studies (SISSA), Trieste, Italy

◉ These authors contributed equally to this work.
* epiasini@sissa.it (EP); zoccolan@sissa.it (DZ)

**Data availability statement:** The data and source code for model fitting and generating figures of this manuscript are available in a publicly accessible repository on Zenodo 10.5281/zenodo.17280352.

## Abstract

The brain combines information from multiple sensory modalities to build a consistent representation of the world. The principles by which multimodal stimuli are integrated in cortical hierarchies are well studied, but it is less clear whether and how unimodal inputs shape the processing of signals carried by a different modality. In rodents, for instance, direct connections from primary auditory cortex reach visual cortex, but studies disagree on the impact of these projections on visual cortical processing. Both enhancement and suppression of visually evoked responses by auditory inputs have been reported, as well as sharpening of orientation tuning and improvement in the coding of visual information. Little is known, however, about the functional impact of auditory signals on rodent visual perception. Here we trained a group of rats in a visual temporal frequency (TF) classification task, where the visual stimuli to categorize were paired with simultaneous but task-irrelevant auditory stimuli, to prevent high-level multisensory integration and investigate instead the spontaneous, direct impact of auditory signals on the perception of visual stimuli. Rat classification of visual TF was strongly and systematically altered by the presence of sounds, in a way that was determined by sound intensity but not by its temporal modulation. To investigate the mechanisms underlying this phenomenon, we developed a Bayesian ideal observer model, combined with a neural coding scheme where neurons linearly encode visual TF but are inhibited by concomitant sounds by a measure that depends on their intensity. This model captured very precisely the full spectrum of rat perceptual choices we observed, supporting the hypothesis that auditory inputs induce an effective compression of the visual perceptual space. This suggests an important role for inhibition as the key mediator of auditory-visual interactions and provides clear, mechanistic hypotheses to be tested by future work on visual cortical codes.

**Funding:** This work was funded by the European Union – NextGenerationEU – PNRRM4C2-I.1.1, in the framework of the PRIN Project no. 2022WX3FM5, CUP:G53D23003220006 (DZ) and PRIN Project no. 2022XE8X9E, CUP:G53D23004590001 (EP). The views and opinions expressed are solely those of the authors and do not necessarily reflect those of the European Union, nor can the European Union be held responsible for them. The study was also supported by the Italian Ministry of University and Research under the call PRO3, project NEMESI (DZ). The funders did not play any role in the study design, data collection and analysis, decision to publish or preparation of the manuscript.

## Author summary

In the brain, information from different senses converges on specialized regions where it is combined in a multimodal representation of the environment. However, inputs from a given modality can directly affect processing in a different modality via direct projections from a given unimodal region (e.g., auditory) to a different one (e.g., visual). Here, we investigated the impact of these auditory-mediated influences on rat visual perception. By combining behavioral experiments with computational modeling, we found that pairing task-irrelevant sounds to drifting gratings systematically altered how these visual stimuli were perceived by the animals, in a way that is consistent with an effective compression of the visual perceptual space induced by the auditory inputs. This, in turn, suggests that inhibition mediates auditory-visual interactions at the neural representation level.

## Introduction

To provide a coherent and reliable representation of the sensory environment, the information collected by our senses needs to be combined in multimodal percepts. The traditional view is that the process of multimodal integration takes place in high-order association cortices, after sensory signals have been independently and extensively processed by unimodal cortical regions. This view is supported by many psychophysical and neurophysiological studies, showing that multimodal representations in such regions often combine unimodal inputs in a statistically optimal way, increasing the accuracy of perceptual judgements [1–12].

More recent work, however, has shown that cross-modal interactions already occur at the level of primary sensory cortices [13]. Imaging studies in humans, as well as single-unit recordings in monkeys have shown that neuronal responses in auditory cortex are modulated by both visual and somatosensory inputs [14–18]. Conversely, sound-driven responses and facilitation of peak latencies under audiovisual stimulation have been reported in human primary visual cortex (V1) [19] and auditory-mediated enhancement of BOLD responses in V1 appears to partially account for the sound-induced flash illusion [20] – the illusory perception of multiple light flashes when a single flash is paired to multiple, closely spaced beep sounds [21–23].

These early cross-modal interactions could be mediated by the direct projections from auditory cortices to V1 that have been reported in human and monkey neuroanatomy studies [24–27]. In primates, however, these projections are sparse, originate mainly from parabelt areas of auditory cortex and target mainly peripheral V1. Denser cortico-cortical connections from primary auditory cortex (A1) to V1 have been instead found in rodents [28–32]. Given the accessibility of these model systems to the genetic dissection of neural circuits [33,34], several authors have investigated the impact of A1 inputs to V1 on visual cortical dynamics and encoding of visual information. Surprisingly, the conclusions of these studies are highly heterogeneous and often contrasting.

Sound was originally reported to hyperpolarize supragranular pyramidal cells in mouse V1, by recruiting a translaminar GABAergic network activated by A1 projections to infragranular V1 neurons [29]. Later studies, however, found that most A1 projections terminate in superficial layers of V1 and engage local inhibitory and disinhibitory circuits underlying a much richer variety of auditory influences on V1: enhancement and suppression of visually evoked responses have both been documented, as well as sharpening of orientation tuning and improvement in the coding of visual information [28,30,35–37]. This heterogeneous

assortment of sound-mediated modulations in V1 seems to depend on factors such as the contrast of the visual stimuli, the luminosity of the environment, the spectral (e.g., pure tones vs. noise bursts) and envelope (e.g., loud vs. quiet onset) properties of the sounds and the temporal congruency between visual and auditory stimuli. Finally, it remains debated the extent to which, in awake mice, sound-driven responses in V1 reflect auditory inputs or, rather, behavioral modulation via sound-evoked orofacial movements [31,35,38].

Amid such contrasting findings at the cortical circuitry level, the functional impact of auditory signals on rodent visual perception has been poorly explored. Does sound improve acuity in visual discrimination tasks? Can any signature of sound-induced suppression (or enhancement) of visual representations be found at the behavioral level? Is the temporal consistency between sounds and visual stimuli important in this regard? The goal of our study was to address these questions, by measuring the extent to which task-irrelevant sounds can affect the perceptual choices of rats engaged in a visual classification task.

## Results

In our study, we trained 12 rats in a visual temporal frequency (TF) classification task, where the visual (V) stimuli to categorize were paired with simultaneous but task-irrelevant auditory (A) stimuli (Fig 1). The visual stimuli were circular sinusoidal gratings (see examples in Fig 1A, 1B) moving outward with 9 possible, randomly selected TFs (Fig 1C; x axis). Rats categorized these stimuli as "Low TF" or "High TF", relative to an ambiguous boundary value (2.12 Hz) in the middle of the tested TF range. The auditory stimuli consisted of either fixed amplitude or amplitude-modulated (AM) white noise bursts (Fig 1D; see also S2 Fig for details). The sounds were uninformative about the correct classification of the visual stimuli, to avoid high-level multisensory integration effects and focus instead on the direct impact of auditory signals on the representation and perception of visual stimuli. Following an approach we have established in previous studies [39,40], the rats had to lick the central response port in an array of three to trigger stimulus presentation and then lick either the left or the right port to choose the "Low TF" or the "High TF" category (with the stimulus-response association being counterbalanced across animals), so as to collect liquid reward in case of correct classification (Fig 1B; see the Materials and Methods for details).

During the training phase, the gratings were paired with a constant amplitude sound - i.e., a white noise burst sampled at 44.1 kHz (Fig 1D, green trace). As explained in the Materials and Methods, rats were gradually introduced to the visual TFs they had to classify, starting from the most discriminable ones, i.e., 0.25 vs. 4 Hz (Fig 2; Training Phase 2). Once they achieved a performance higher than 70% correct in at least two consecutive sessions, gratings with additional TFs were progressively added through Training Phases 3-5, until, in Training Phase 5, the animals had to classify the full range of visual TFs. Responses to the 2.12 Hz boundary stimulus were randomly rewarded. Two of the rats failed to achieve the performance criteria in Training Phase 2, after a total of 28 sessions in Phase 1 and 2, and were therefore excluded from further testing. The contrast of the gratings was initially set at 100%, but it was progressively reduced to 25% (Fig 2; gray patches), given that the impact of sound on V1 responses has been found to be stronger when visual stimuli are shown at low/intermediate contrast [28,35]. As illustrated in Fig 2 for two example rats, the contrast reduction procedure was applied several times, to ensure that the animals would successfully classify the gratings' TFs at both the lowest and full contrast. More specifically, our procedure included: 1) an early exploratory stage, where contrast was quickly reduced, within 6-10 sessions, from 100% to values as low as 15% (e.g., see sessions 39-40 in Fig 2A, top); 2) a re-exposure to full contrast in case the performance had dropped to chance following a step of

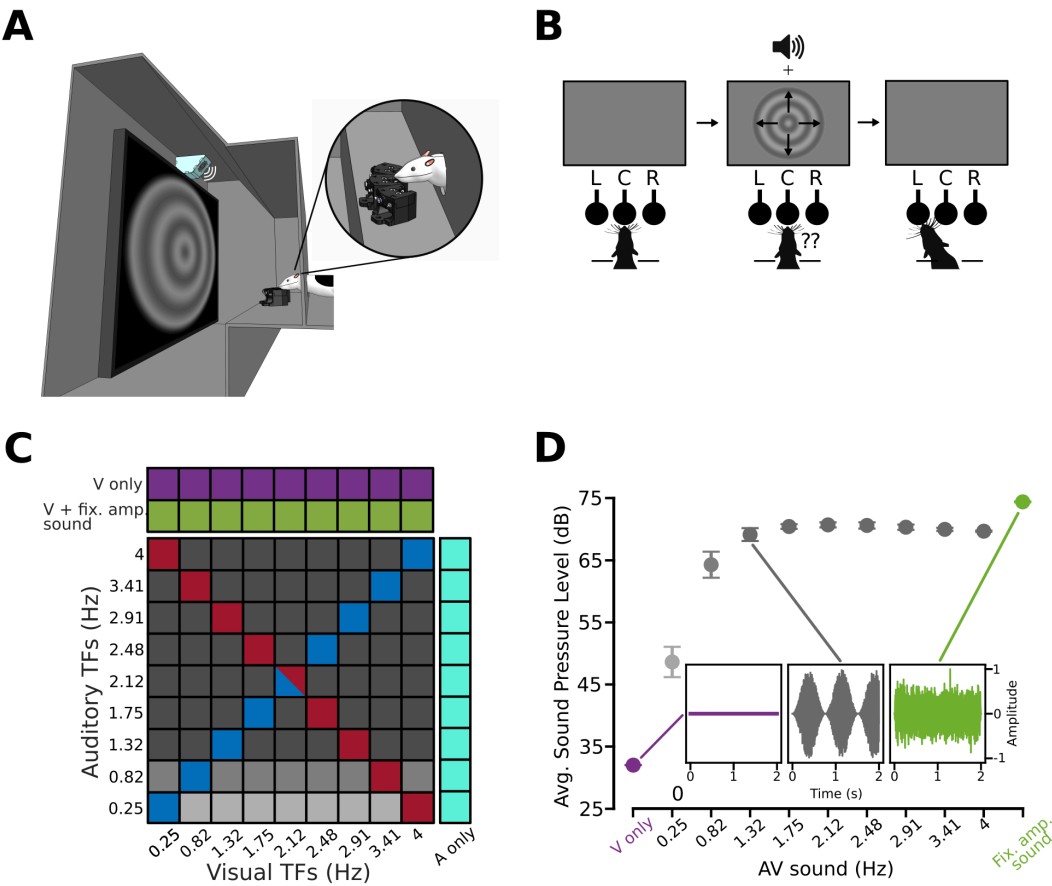

**Fig 1. Experimental setup and audio-visual stimulus conditions. *A*.** Rendering of a rat extending its head through a viewing hole in the wall of a sound-isolated operant box, equipped with a display and a speaker for presentation of the audio-visual stimuli. The hole was placed in front of the stimulus display and the speaker (in cyan). The hole also allowed access to three nose-poke ports used to trigger stimulus presentation and report the temporal frequency of the visual stimuli. ***B*.** Schematic representation of a behavioral trial. As in previous studies [39,40], the rats learned to protrude their head through the viewing hole and trigger stimulus presentation by touching the central nose-poke port. The visual stimuli were circular, outward-moving sinusoidal gratings that rats classified according to their temporal frequency (TF). Gratings with TF < 2.12 HZ had to be classified as "Low TF"; those with TF > 2.12 Hz as "High TF". Each class was associated with one of the two lateral nose-poke ports. Auditory stimuli did not provide any information about the correct classification of the visual stimuli. ***C*.** Stimulus conditions presented to the rats during the test phase of the experiment. These included: 1) visual gratings paired with fixed amplitude sounds (green); 2) visual gratings paired with AM sounds (cells of the square matrix: blue, gray and red refer, respectively, to congruent, incongruent and anti-congruent audiovisual conditions; the shades of gray, from light to dark, indicate AM sounds with progressively larger intensity; see Methods and S2 Fig for details); 3) unimodal purely visual gratings (purple); and 4) unimodal purely auditory AM sounds (cyan). ***D*.** Details on the auditory stimuli. Green dot: white noise burst with a fixed maximal amplitude. Gray dots: amplitude-modulated white noise bursts, each with a different temporal frequency of the sinusoidal envelope. Purple dot: absence of an auditory stimulus. See Methods and S2 Fig for details on estimation of average perceived sound intensity. Error bars: s.e.m. across rats. For the modulated noise bursts, different shades of gray are used to group stimuli based on their average sound intensity (same as in ***C***). Insets: sound amplitude control signal for white noise burst (green), the absence of auditory stimulus (purple) and one example of modulated noise burst (gray).

contrast reduction (see sessions 40-46, following the accuracy drop in session 39, in Fig 2B, top); and 3) one last re-exposure to full contrast before the final, stable reduction to 25%. Rats moved quickly through the training phases, occasionally displaying some drop of discrimination accuracy after transitioning from one phase to the next, which, in some cases, required

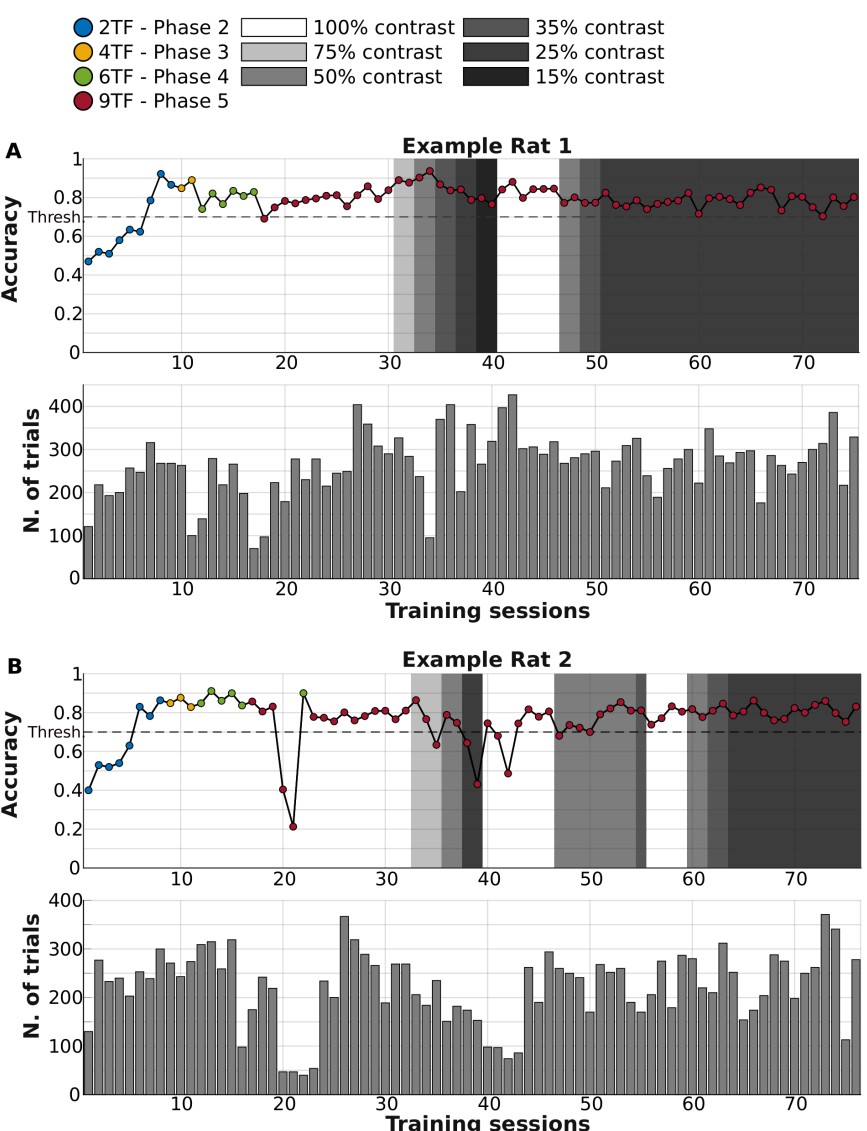

**Fig 2. Discrimination accuracy of two example rats across the different phases of the training procedure.** *A.* The line plot (top) shows the percentage of correct choices of Example Rat 1 in classifying the temporal frequency of the drifting gratings across more than 70 daily training sessions. The color of the dots indicates the training phase (from 2 to 5; see caption), while the intensity of the gray patches denotes the various stages of contrast reduction that were performed during the training (see legend and the Materials and Methods). The dashed horizontal line represents the criterion accuracy (70%) required to progress to the next training phase. To advance, rats had to achieve performance above criterion for at least two consecutive days within the same training phase. The bar plot (bottom) shows the number of behavioral trials that were collected for the rat in each daily session. *B.* Same as *A*, but for Example Rat 2.

moving them back to the previous phase (e.g., Example Rat 2 went back to Phase 4 for one session after a substantial performance drop following 5 sessions in Phase 5; see Fig 2B). These temporary drops, lasting a few sessions, are typical for rodents that are consolidating the acquisition of a perceptual task [41] and are likely due to fluctuations in the level of motivation, deployment of exploratory strategies [42], or temporary failures of the behavioral rig (e.g., uneven delivery of liquid from the reward ports), as suggested by the often concomitant

decrease in the number of behavioral trials completed by the animals (see Fig 2B, bottom). Nevertheless, all the animals eventually achieved the criterion performance of 70% correct choices with 25% contrast gratings that was necessary to advance them to the test phase.

### Task-irrelevant sounds affect rat perception of drifting gratings in a way that is independent of the temporal congruency between the auditory and visual stimuli

Once the animals had become proficient in the TF classification task with 25% contrast gratings, they were moved to the testing phase, where they faced a rich variety of audiovisual stimuli (Fig 1C). The gratings were paired not only with the fixed amplitude burst used during training (Fig 1C, green cells), but also with AM white noise stimuli. These sounds were obtained by modulating the amplitude of a white noise burst with sinusoids having 9 possible TFs (the same of the visual gratings). The rats were presented with all possible pairwise combinations of visual and auditory TFs (cells in the squared matrix of Fig 1C). This yielded trials where the auditory and visual stimuli had either the same TF (blue cells) or different TFs (gray and red cells). The latter included a special pool of anti-congruent conditions (red cells), where progressively larger visual TFs were paired with progressively smaller auditory TFs. Additionally, we also tested unimodal, purely visual trials (purple cells) and unimodal, purely auditory trials with AM sounds (cyan cells). Since the rats were never trained to classify the TFs of the auditory stimuli, with the latter trials we simply measured the spontaneous responses of the animals to the AM sounds, without providing any feedback about their choices (see Materials and Methods). The rats did not spontaneously classify the AM bursts according to their TF in the auditory only conditions, thus confirming the task-irrelevance of the sounds (see S1 Fig).

Overall, this design allowed probing the impact of task-irrelevant auditory stimuli on visual perception along two distinct dimensions: 1) whether sound alters the sensitivity of visual discrimination or, rather, introduces a bias in visual classification; and 2) whether the impact of sound depends on the temporal consistency of audio-visual stimuli or, rather, on the intensity of the auditory stimuli.

When the gratings were paired with the fixed amplitude noise, the group average psychometric curve reporting the proportion of "High TF" choices as a function of the grating frequency was sharp and symmetrical, with the point of subjective equality sitting squarely on the ambiguous TF (Fig 3A, green). Interestingly, pairing the gratings with either the congruent (blue) or anti-congruent (red) AM sounds produced an identical increase in the proportion of "High TF" choices, which was particularly prominent for the highest TFs. This observation was confirmed by a 2-way ANOVA having as factors the audiovisual experimental condition and the grating TF (see the Materials and Methods). Both factors significantly modulated rat choices and so did their interaction (Table 1), thus confirming that the increase of "High TF" choices yielded by the modulated sounds was not homogeneous along the TF axis. A post-hoc Tukey test confirmed that the curves observed for the two conditions featuring AM sounds were statistically indistinguishable. This indicated that the congruency between the TF of the AM sounds and the TF of the gratings did not affect the impact of the auditory stimuli on the visual discrimination task.

This observation led us to group together all the trials where the visual stimuli were paired with an AM sound, regardless of the congruency of their TFs. The resulting psychometric curve (Fig 3B, black) featured an asymmetrical vertical shift, with respect to the reference curve measured with the fixed amplitude sounds (green dots/line), that was equivalent to the one previously observed with the congruent and anti-congruent AM sounds (compare

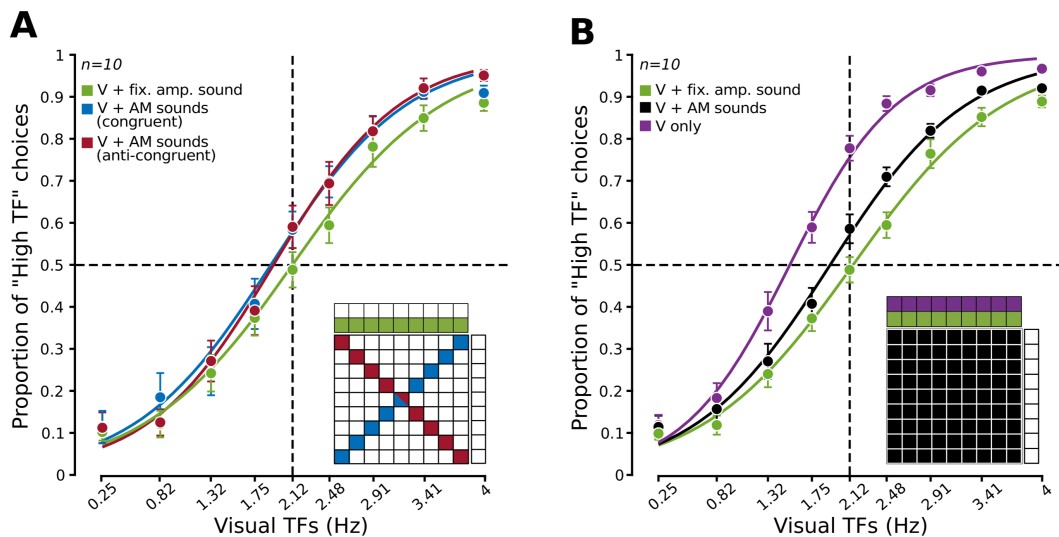

**Fig 3. Task-irrelevant sounds strongly modulate rat classification of the temporal frequency of drifting gratings.** *A*. Group average proportion of "High TF" choices (*n*=10 rats) as a function of the TF of the visual gratings, when the latter were paired with: 1) fixed amplitude sounds (green); 2) congruent AM sounds (blue); and 3) anti-congruent AM sounds (red). The error bars denote s.e.m. over the group of 10 rats. The curves are the result of logistic regressions to the data points. The inset illustrates which of the stimulus conditions shown in Fig 1C contributed to the data points/curves. See Table 1 for a statistical comparison among the three curves. *B*. Same as in *A*, but for stimulus conditions where the gratings were paired with: 1) fixed amplitude sounds (green; same data as in *A*); 2) AM sounds (black); and 3) no sounds (purple). See Table 2 for a statistical comparison among the three curves.

**Table 1. Two-way ANOVA table (A) and Tukey post-hoc tests (B) for the comparison among rat psychometric curves shown in Fig 3A.** A repeated-measures two-way ANOVA was run on the proportion of "High TF" choices reported in Fig 3A, having as factors the experimental condition ("V + fixed amplitude sound", "V + AM congruent" and "V + AM anti-congruent") and the grating TF.

| (A) Two-way ANOVA with repeated measures | | | | | |
|---|---|---|---|---|---|
| Effect | df | SS | MS | F | p |
| Visual TF | 8 | 33.483 | 4.185 | 289.291 | $0.001 \cdot 10^{-48}$ |
| Experimental condition | 2 | 0.212 | 0.106 | 4.897 | 0.02 |
| Visual TFs:Experimental condition | 16 | 0.191 | 0.012 | 2.533 | 0.0018 |
| (B) Tukey post-hoc tests | | | | | |
| Condition 1 | Condition 2 | Difference | StdErr | p | Lower | Upper |
| V + AM sounds (congruent) | V + AM sounds (anti-congruent) | −0.0069 | 0.0118 | 0.8297 | −0.0397 | 0.0259 |
| V + AM sounds (anti-congruent) | V + fixed-amplitude sound | 0.0625 | 0.0254 | 0.0839 | −0.0084 | 0.1336 |
| V + fixed-amplitude sound | V + AM sounds (congruent) | −0.0556 | 0.0256 | 0.1298 | −0.1271 | 0.0159 |

to the blue and red curves in Fig 3A). Strikingly, a similar but even more prominent shift of the psychometric curve was observed for the unimodal visual conditions, when the gratings were presented without any concomitant sound (purple dots/line). Again, a two-way ANOVA showed a significant main effect of experimental condition, TF of the gratings and their interaction, and a post-hoc Tukey test revealed a significant difference between the curve obtained with the unimodal visual stimuli and the curves measured with the paired sounds, either AM or fixed amplitude (Table 2).

These results showed that rat perception of visual temporal frequencies was systematically shifted towards reporting the "High TF" category if the noise bursts became amplitude-modulated and, even more so, when the sounds disappeared altogether. Wondering about the possible cause of these progressively larger shifts, we realized that one key attribute of

**Table 2**. **Two-way ANOVA table (A) and Tukey post-hoc tests (B) for the comparison among rat psychometric curves shown in Fig 3B**. A repeated-measures two-way ANOVA was run on the proportion of "High TF" choices reported in Fig 3B, having as factors the experimental condition ("V + fixed amplitude sound", "V + AM sound" and "V only") and the grating TF.

| (A) Two-way ANOVA with repeated measures | | | | | |
|---|---|---|---|---|---|
| Effect | df | SS | MS | F | p |
| Visual TF | 8 | 33.809 | 4.226 | 314.013 | $0.002 \cdot 10^{-53}$ |
| Experimental condition | 2 | 1.766 | 0.883 | 32.593 | $0.001 \cdot 10^{-5}$ |
| Visual TFs:Experimental condition | 16 | 0.461 | 0.29 | 8.763 | $0.001 \cdot 10^{-11}$ |
| (B) Tukey post-hoc tests | | | | | |
| Condition 1 | Condition 2 | Difference | StdErr | p | Lower | Upper |
| V + fixed-amplitude | V + AM sounds | −0.0614 | 0.0244 | 0.0769 | −0.130 | 0.0067 |
| V + AM sounds | V only | −0.1324 | 0.1776 | 0.0001 | −0.182 | −0.0829 |
| V only | V + fixed-amplitude | 0.1938 | 0.0299 | 0.0003 | 0.1102 | 0.2773 |

the sound did decrease from the fixed amplitude to the AM bursts and then further to the purely visual conditions: its intensity. In fact, because of the sinusoidal modulation, there were moments during the AM bursts where the instantaneous sound intensity dropped to zero (S2B Fig). As a result, the average sound intensity (as computed over the inferred reaction time of the animals) was lower for the AM bursts than for the fixed amplitude burst (see Figs 1D and S2C), besides being minimal for the purely visual stimuli (corresponding, in this case, to the environmental background noise). In a previous study [29], noise bursts were reported to hyperpolarize V1 neurons, with the magnitude of this inhibition increasing monotonically with their intensity. Given this observation, we hypothesized that the perceptual effect of sound is to compress visual cortical representations in a way that is proportional to its intensity.

## A computational model accounts for the behavioral impact of task-irrelevant sounds as the result of visual perceptual space compression

To test the hypothesis that response suppression is the key mediator of auditory influences on visual representations, we developed a Bayesian ideal observer model of rat perceptual choices. The model rests on three assumptions, which are illustrated in the cartoons of Fig 4 (the full mathematical derivation of the model can be found under the "Ideal observer model" section in the Materials and Methods). The first is that V1 neurons respond to progressively larger TFs with an approximately linear increase of firing rates, at least within the TF range tested in our study (Fig 4A). Although no systematic studies exist about the tuning of rat visual cortical neurons for temporal frequency, this hypothesis is grounded on the periodic activity bursts evoked in rat V1 simple cells by the succession of luminance waves in drifting gratings [43–45]. It is therefore reasonable to assume that gratings with a larger number of cycles (i.e., higher TF) will tend to drive V1 simple cells more vigorously. The second assumption is that sound dampens neuronal responses, thus lowering the slope of the linear dependency of neuronal firing from the TF of the gratings, with this decrease being stronger for higher intensity sounds (Fig 4A; compare the purple line, corresponding to the unimodal, purely visual condition, to the gray and green lines, corresponding to conditions where the visual stimuli are paired with sounds with progressively larger intensity). This hypothesis is consistent with the sound-induced (and intensity-dependent) hyperpolarization of V1 pyramidal neurons reported by [29], as well as with the inhibitory impact of sounds on V1 responses reported by several authors under certain stimulus conditions [30,35,36] (see the Discussion for details). In the representational space defined by the activity of a population of V1 neurons, such response suppression results in a compression of the population vectors

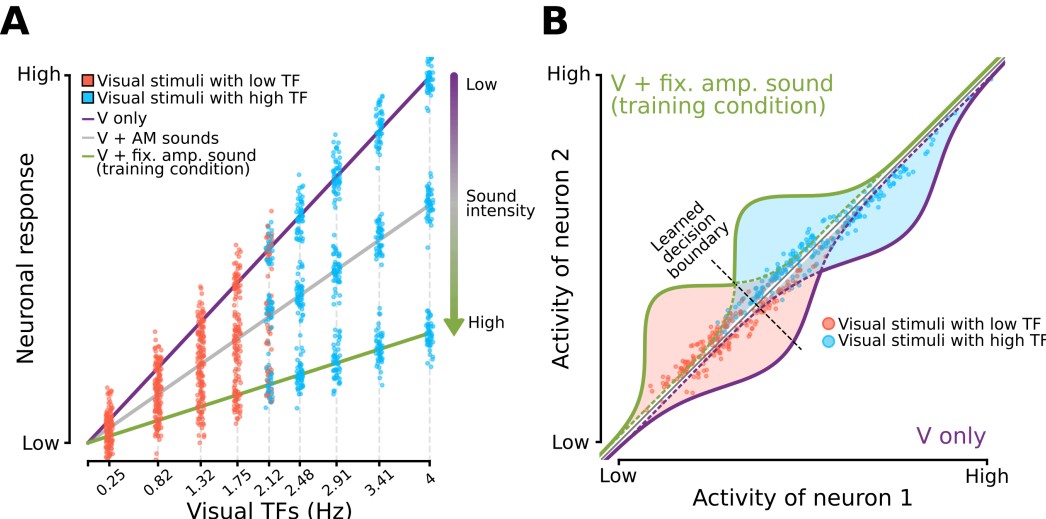

**Fig 4. Schematic of an ideal observer model where neuronal populations encode linearly visual temporal frequencies and are suppressed by concomitant sounds in a way that depends on their intensity.** *A*. Hypothesized tuning of a visual cortical neuron in the ideal observer model. The neuron responds to the visual gratings with an average firing rate that grows linearly as a function of the grating TF. Due to neuronal noise, the responses to repeated presentations of the same grating are distributed according to a Gaussian function centered on the average firing rate elicited by the grating. More specifically, the red and cyan dots refer to those responses that are elicited by the gratings that, in our task, had to be classified as belonging to the "Low TF" and "High TF" classes, respectively. When only the gratings are shown (purely visual condition), the neuronal response evoked by any given TF reaches its maximal possible value (purple line). When the gratings are paired with concomitant sounds, the responses to all the TFs are inhibited by a factor that depends on sound intensity, resulting in a reduction of the slope of the linear dependency between firing rate and TF. Since the AM sounds have a lower average intensity than the fixed amplitude sound (see Fig 1D, Materials and Methods, and S2 Fig), the resulting compression of the firing rate is larger for the latter (green line) than for the former (gray line). *B*. Illustration of the ideal observer model when a population of multiple neurons is considered. The drawing shows a hypothetical neuronal representational space made of two units. Each neuron behaves as shown in *A*. In the example, two TFs yields two distributions of population vectors (red and cyan dots). When the gratings are paired with a sound with strong intensity (e.g., the fixed amplitude noise burst), all responses are inhibited and, as a result, the distributions are compressed toward the origin of the representational space (green curves). Since rats are trained to discriminate the gratings under these conditions, they learn a decision boundary (dashed line) that optimally separates such compressed distributions. When the animals are presented with unimodal, purely visual gratings, the sound-induced inhibition is released and the response distributions shift towards higher firing rate values (purple curves). The animals, however, still rely on the previously learned decision boundary to discriminate these stimulus conditions. As a result, the overall proportion of "High TF" choices increases (i.e., more red dots fall on the "false alarm" side of the decision boundary).

encoding the visual TFs towards the origin of the space (Fig 4B; compare the green to the purple distributions). Finally, the third hypothesis underlying our model is that downstream decision neurons learn an optimal discrimination boundary in the representational space (Fig 4B; dashed line) under the sensory conditions they were repeatedly exposed to during training (i.e., with the visual gratings being paired with the fixed amplitude sound; green distributions in Fig 4B). This optimality is what gives our model its character as an ideal observer [46,47]. We also assume that the boundary remains unchanged when new conditions (e.g., the purely visual gratings; purple distributions in Fig 4B) are interleaved with those used for training.

In our model, on each trial, the combination of a visual and an auditory stimulus generates an internal representation sampled from a Normal distribution with fixed standard deviation $\sigma$, centered on some average representation of the TF value of the visual stimulus $s$. Following from the assumptions above, the internal representation is then scaled by a factor $\gamma$ (see the Ideal observer model section in the Materials and Methods), and the decision rule is the

one that would be Bayes-optimal in the training condition (i.e., with the gratings paired with the fixed amplitude sound). We can thus derive the probability for a rat of classifying a given stimulus *s* as belonging to the *H* (i.e., "High TFs") category as

$$p(\text{report } H \mid s) = \Phi\left[\frac{\gamma s - s_0}{\sigma}\right],\tag{1}$$

where $\Phi$ is the standard Normal cumulative function and $s_0$ is the boundary 2.12 Hz visual stimulus that separates the "Low TFs" class from the "High TFs" class (see the Ideal observer model section in the Materials and Methods). In the model, the $\sigma$ parameter controls the slope of the psychometric function and measures the internal noise of the representation (i.e., rat sensitivity over the visual TF axis), while the $\gamma$ parameter controls the gain of the internal perceptual representation.

This model allowed testing alternative hypotheses about the impact of sounds on visual representations, depending on the number of distinct values that the $\sigma$ and $\gamma$ parameters were free to take. At one extreme, allowing a different value of $\sigma$ for each sound condition would represent the hypothesis that sound affects the sensitivity of the visual representation (i.e., the sharpness of the psychometric curves). At the other extreme, keeping a constant $\sigma$ but allowing a different value of $\gamma$ for each level of sound intensity would represent the hypothesis that sound compresses the visual representation according to its intensity, but leaves visual perceptual sensitivity unaltered.

Using the model to test for the impact of sound intensity on visual perception required estimating the sound intensity experienced by the rats in the time interval between stimulus onset and each animal's estimated reaction time (see the Materials and Methods and S2B Fig, second to last row). Analyzing the resulting, trial-averaged intensity levels revealed a finer-grain grouping of stimulus conditions (see Figs 1D and S2C) compared to the one shown in Fig 3B. In addition to the conditions with maximal and minimal intensity (i.e., 74.8 dB for the fixed amplitude sounds and 32 dB for the purely visual stimuli, respectively), we found that the AM conditions could be grouped into three main clusters with increasingly larger intensity, based on the TF at which the noise bursts were modulated: TF = 0.25 Hz (48.5 $\pm$ 2.4 dB; mean $\pm$ s.e.m. across rats); TF = 0.82 Hz (64.2 $\pm$ 2.3 dB); and 1.32 Hz $\leq$ TF $\leq$ 4 Hz (70.7 $\pm$ 0.5 dB; these conditions are shown with progressively darker shades of gray in the stimulus matrix of Figs 1C, 1D and S2).

We inferred the parameters of the ideal observer using a hierarchical Bayesian approach (see the section Ideal observer model of the Materials and Methods, S3 and S4 Figs for details on the model and our inference approach), which yielded a posterior probability distribution over the values $\sigma$ and $\gamma$ that characterized each specific rat (referred to as rat-level parameters), as well as the values that characterized all rats taken together (referred to as group-level parameters). The group-level estimates obtained for a model with a single $\sigma$ and 5 different $\gamma$ per rat (i.e., one $\gamma$ for each of the 5 levels of sound intensity in our stimulus set) are summarized by the psychometric curves shown in Fig 5A.

The curves fitted tightly the measured classification accuracies and captured well the way the proportion of "High TF" choices progressively increased, while sound intensity dropped from maximal (green curve; fixed amplitude sound) to minimal (purple curve; no sound), passing through the three intermediate levels in which the AM sounds were clustered (gray curves) (Fig 5; see Materials and Methods and S5 Fig for a posterior predictive analysis illustrating the tightness of the model fit to the data, and S1 Table for details on fitted model parameters and inference diagnostics). This means that an ideal observer with constant sensitivity (i.e., a single $\sigma$) could describe well the impact of sound on rat visual perceptual choices,

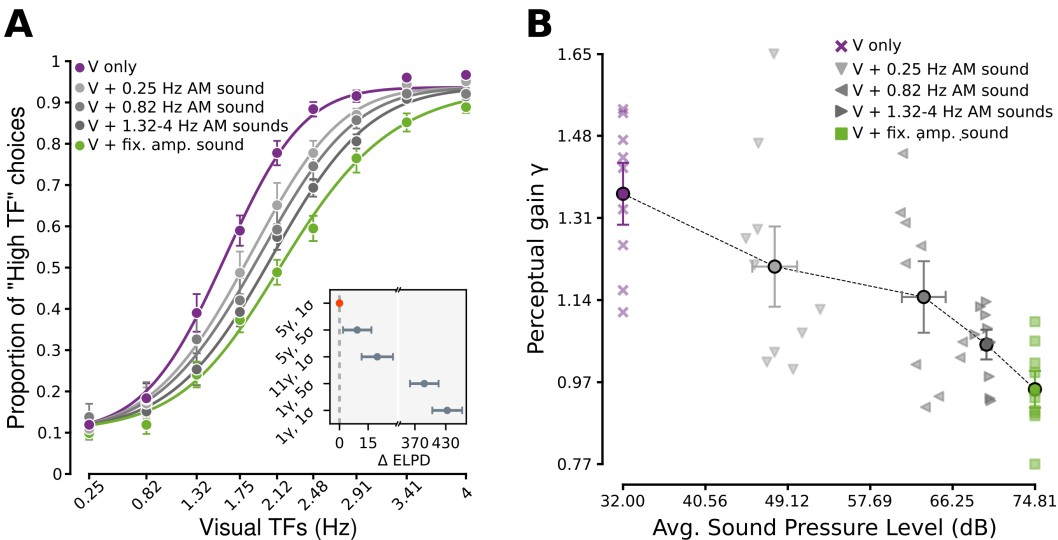

**Fig 5. An ideal observer model where task-irrelevant sounds compress rat visual perceptual space according to their average intensity accounts for rat perceptual choices. *A*.** Group average proportion of "High TF" choices (*n*=10 rats) as a function of the TF of the visual gratings, when the latter were paired with: 1) fixed amplitude sounds (green); 2) AM sounds with increasingly larger TFs (progressively darker shades of gray, matching those in the stimulus matrix shown in Fig 1C); and 3) no sounds (purple). The error bars denote s.e.m. over the group of 10 rats. The curves are group-level predictions of the proportions of "High TF" choices according to the Bayesian ideal observer model defined in Eq (1), with a single value of the sensitivity parameter $\sigma$ and 5 different values of the scaling factor $\gamma$, one for each level of sound intensity (more precisely, each curve is the mean of the distribution of psychometric curves induced by the posterior distribution of the $\sigma$ and $\gamma$ parameters obtained for a given sound intensity level). For this model and the data as presented in the plot, $R^2 = 0.9973 \pm 0.0002$ (posterior mean $\pm$ st. dev.). The inset shows the difference between the expected log predictive density (ELPD) of the reference model with 1 $\sigma$ and 5 $\gamma$ values (i.e., the one yielding the psychometric curves) and variants of the model having a given number of free $\sigma$ and $\gamma$ parameters (as indicated on the ordinate axis). In all comparisons, the reference model was the one with lowest $\Delta$ELPD, i.e., with the highest predictive power. *B*. Relationship between the magnitude of the scaling factor $\gamma$ in the ideal observer model and the intensity of the sounds that were paired with the visual gratings. Triangle, square and cross symbols refer to individual rats, while circles are group averages. Sound conditions are labeled according to the same color code as in *A* and Fig 1C. Vertical error bars denote the std. dev. of the posterior distribution of the group-level $\gamma$ parameters (see Methods). The horizontal error bars denote s.e.m. over the group of 10 rats.

provided that the internal representation of visual TFs was appropriately scaled by a gain factor ($\gamma$). Importantly, the magnitude of this gain decreased monotonically across the 5 levels of increasing sound intensity in which our stimulus conditions could be grouped (Fig 5B).

To verify that this was the best description of the data, we compared the predictive power of this "reference" model to that of other variants, with a different number of values that $\sigma$ and $\gamma$ were allowed to take. The predictive power of each model was measured using the expected log predictive density (ELPD), that is, the average, expected log-likelihood of the model on new data. This metric is estimated by leave-one-out cross-validation using a standard approach and automatically penalizes overly complex models that tend to overfit (see section Model comparison in the Materials and Methods). In our comparison, we computed the difference in ELPD between the reference model with 1 $\sigma$ and 5 $\gamma$ and each other model. We refer to this metric as $\Delta$ELPD.

The reference model with a fixed $\sigma$ and 5 $\gamma$ (one for each of the 5 levels of sound intensity) was the best (i.e., the one with the lowest $\Delta$ELPD) of a broad range of models we tested (Fig 5A, inset). In particular, allowing a separate $\gamma$ for each of the 9 AM conditions, thus obtaining a total of 11 possible $\gamma$ values (while still keeping $\sigma$ fixed to a single value), yielded a worse model (an indication of overfitting). This confirmed that noise bursts modulated within

the [1.32 Hz, 4 Hz] range were all equivalent in terms of their impact on visual perceptual choices, as expected given their very similar intensity (see Figs 1D and S2). A model with 5 $\sigma$ and 1 $\gamma$ also performed much worse than the reference model, thus confirming that the different sound stimuli did not impact rat sensitivity to the visual TFs, but, rather, the dynamic range of the internal representation of the TFs. This conclusion was further reinforced by noticing that not even a model with 5 $\sigma$ and 5 $\gamma$ outperformed the reference model with only 1 $\sigma$ and 5 $\gamma$. Finally, in order to describe more quantitatively the relationship between perceptual gain and sound intensity, we also compared the models just described with one that puts a linear constraint on the dependence between these two quantities (while keeping $\sigma$ fixed as in the best-performing model seen so far). This model performed just as well as the $(5\gamma, 1\sigma)$ model, confirming quantitatively that $\gamma$ decreases for larger values of sound intensity (slope: $(-8.6 \pm 1.4) \times 10^{-3}$, intercept: $1.636 \pm 0.083$, posterior mean $\pm$ standard deviation; S6 Fig).

In summary, our Bayesian ideal observer analysis supports the hypothesis that the effect of sound on the representation of temporally modulated visual gratings is to compress the visual perceptual space rather than increase or decrease perceptual uncertainty. Mechanistically, this finding suggests that sound acts as a powerful modulator of the responses of visual cortical neurons (see Fig 4) but leaves their tuning for temporal frequency unaltered.

## Discussion

The goal of our study was to understand the functional impact of task-irrelevant sounds on rodent visual perception. In designing our experiments, we took inspiration from the influences that sounds or direct activation of A1 neurons (e.g., via optogenetics) have been reported to exert on visual cortical activity in rodents. As already mentioned, these influences are extremely heterogeneous.

Iurilli and colleagues [29] found a sound-induced hyperpolarization of V1 L2/L3 pyramidal neurons, suggesting that the functional impact of A1 projections to V1 is mainly the inhibition of visual cortical activity. The authors also showed that pairing a sound with the light flash used to trigger a visually-driven fear response significantly reduced its magnitude, suggesting a degraded processing of the visual stimulus. These findings are consistent with a view where primary sensory cortices engage in mutual interareal inhibition to compete for the delivery of the most behaviorally salient information to downstream decision centers, in agreement with the predominantly suppressive effect of hetero-modal stimuli that has been reported in auditory and somatosensory cortices of monkeys, cats and ferrets [14,16,48–50]. This does not exclude that sound-induced inhibition could differently affect distinct neuronal subpopulations. For instance, it could impact more on the responses of pyramidal cells that are only weakly activated by visual stimuli, thus potentially sharpening the selectivity of V1 neurons. This hypothesis is consistent with a later mouse study [28], showing that orientation tuning of V1 L2/L3 pyramidal neurons was sharpened in the presence of sound, thanks to a disinhibitory circuit activated by A1 projections to V1 interneurons in L1 and resulting in an increase of firing at the preferred orientation of the pyramidal cells. Such a sharpening of stimulus selectivity was confirmed by [37], who showed how pure tones enhanced the responses of V1 neurons tuned for the orientation/direction of the test grating, while suppressing the activity of neurons with orthogonal/opposite preference. The gating of A1 inputs to V1 by L1 interneurons was confirmed by [30], who, however, found the resulting modulation of L2/L3 neurons to be strongly context-dependent – inhibition was dominant in the dark, while excitation prevailed in the light. In addition, this study found that loud sound onsets increased the response of L2/L3 neurons to temporally coincident visual stimuli. This

suggests a functional role of A1 projections in boosting the saliency of visual events that co-occur with abrupt sounds, in agreement with the observed enhancement of BOLD responses in human V1 during the sound-induced flash illusion [20].

Yet another study [36] stressed the dependence of auditory modulation of V1 responses from the features that define the visual and auditory stimuli: visual contrast, sound envelope, and temporal congruency. Again, both sound-mediated enhancement and suppression of visually evoked responses were observed in V1, with suppression becoming dominant when visual drifting gratings were paired with frequency modulated sounds having an incongruent TF—an indication that V1 neurons may encode the temporal congruency between audio-visual stimulus features. More recently, [35] found that V1 responses to drifting gratings were enhanced by the concomitant presentation of auditory white noise, resulting in an increased discriminability of grating orientation and direction from neuronal signals. Interestingly, these auditory modulations were not accounted for by sound-evoked orofacial movements and locomotory behavior, as previously suggested by [38]. The dissociation of early auditory-related and a later motor-related modulations of V1 activity was confirmed by [31], which, however, found that the encoding of visual features was poorly affected by both kinds of influences.

In our study, we designed a discrimination task and a set of audio-visual stimuli that allowed looking for the perceptual manifestation of several of the (often contrasting) neurobiological processes reviewed above. Consistently with [36], our visual stimuli were circular drifting gratings spanning a range of TFs (Fig 1A and Fig 1B), i.e., rich spatiotemporal patterns that are known to strongly activate rat visual neurons [43–45,51]. Also as in [36], some of the sounds that we paired with the gratings were modulated at the same TFs of the gratings in such a way to yield consistent, inconsistent or anti-consistent matches (Fig 1C). However, to maximize the possible enhancement of visual perceptual events by the concomitant sounds, the auditory stimuli were not modulated in frequency but in amplitude (S2A Fig). The rationale was that the periodic bursts of visual cortical activity evoked by the succession of luminance waves in the gratings (as expected, for instance, for V1 simple cells; see [43–45]) could be boosted by the close succession of up-ramps in the sinusoidal envelopes of the AM sounds [30], thus potentially improving the discriminability of the gratings in the case of congruent TFs. At the same time, the amplitude modulation at different TFs yielded sounds with variable average intensity over the response window of the animals (see Figs 1D and S2C). This allowed testing whether the impact of sound on visual perception depended on sound intensity, as expected according to [29].

Overall, our experiments provided a clear answer to the question of whether task-irrelevant sounds have an impact on visual perception, raising specific, testable hypotheses about the nature of the underlying audio-visual interactions.

First, we found that sounds have an impact on rat visual perceptual decisions (compare the purple to the green curve in Fig 3B) as large as that typically reported in multimodal integration studies [1–3]. This means that the direct, automatic influence of auditory inputs on visual representations is as effective on rat perceptual choices as the process of learning how to combine different sensory cues in supra-modal, Bayes-optimal representations.

Second, both the magnitude and direction of auditory influences on rat visual perception were independent of the temporal consistency between the AM sounds and the gratings (compare the red to the blue curve in Fig 3A). This indicates that, differently from what observed by Meijer and colleagues [36] with frequency modulated sounds, no encoding of the congruency between the temporal features of the audio-visual stimuli took place under our experimental conditions.

Third, no sharpening of the psychometric curves was observed – rather, we found an asymmetrical (i.e., TF-dependent) bias towards reporting more often the "High TF" class (Fig 3A). This bias points to a sound-mediated increase of the "High TF" evidence encoded by the visual cortical representation when the AM sounds, rather than the fixed amplitude burst, were paired with the gratings. Assuming that visual cortical neurons encode progressively higher visual TFs with increasingly larger firing rates (see Fig 4A), such enhancement of "High TF" evidence could be accounted for by two alternative mechanisms: excitation or disinhibition of visually driven responses. The first hypothesis would be consistent with the conclusion of Deneux and colleagues [30] that sound onsets are especially effective at boosting visual cortical responses. As such, the succession of up-ramps in the sinusoidal envelopes of the AM sounds would have been more effective in enhancing V1 responses, as compared to the stationary, fixed amplitude noise burst. Hence, the increase of "High TF" evidence carried by the neuronal population and the resulting increase of "High TF" choices. Such a direct excitation of visual cortical neurons would also be consistent with the early cross-modal integration mechanisms that are though to contribute to the sound-induced flash illusion [20–23].

This explanation, however, is not supported by the observation that the proportion of "High TF" choices further increased when sounds were removed altogether (Fig 3B; purple curve). Hence the intuition that sounds must have inhibited, rather than excited, the firing of the visual neurons encoding the TF of the gratings, in a way that was dependent on their intensity (Fig 4A), in agreement with the findings of Iurilli and colleagues [29]. According to this hypothesis, the observed shifts in the psychometric curves were the result of an increasingly large disinhibitory effect when the sounds paired with the gratings progressively reduced their intensity, passing from the fixed amplitude burst to the AM sounds and, finally, to the purely visual conditions. Moreover, according to this hypothesis, AM sounds with lower TF, given their lower average intensity, should have yielded a larger disinhibitory effect and, therefore, a larger shift of the psychometric curve, as compared to AM sounds with higher TF. This trend was indeed observed in our data (Fig 5).

Importantly, the above-mentioned hypothesis was implemented in a Bayesian ideal observer model of rat perceptual choices, combined with a simple neural coding scheme. The scheme assumes the existence of a population of visual neurons that encode the TF of the gratings with linearly increasing firing rates and that are inhibited by sounds by a measure that depends on sound intensity (Fig 4). These assumptions, along with the further assumption of a fixed decision boundary learned by the rats during training, led to a model (Eq. 1) that was able to capture very precisely the full spectrum of psychometric curves we measured, using just one parameter for rat sensitivity over the TF axis and five levels of sound-induced suppression (Fig 5A, inset), related to sound intensity (Fig 5B).

Overall, these findings imply that, among the variety of auditory influences that have been reported in rodent visual cortex [28–31], inhibition plays the dominant role at the functional level, at least under the experimental conditions of our study. This conclusion is in agreement with the findings of Iurilli and colleagues [29]. It also resonates with the experience-dependent suppression of V1 activity by auditory stimuli reported by Garner and Keller [52]. This study, however, is not directly comparable to ours because: (1) audio and visual stimuli were delivered sequentially and not concomitantly; and (2) auditory stimuli were always informative about the upcoming delivery of a reward, unlike in our task where they were completely devoid of information.

At the same time, our results appear inconsistent with the sound-induced enhancement of visual cortical responses and the sharpening of feature selectivity reported by some authors

[28,35,37]. One possibility to reconcile these findings is that auditory signals may differently affect distinct kinds of perceptual tasks. In tasks requiring the discrimination of visual features that are related to the energy of the stimulus (such as its TF) and that are likely encoded by locally linear tuning curves (as postulated in our model; see Fig 4A), sound-mediated engagement of inhibitory circuits may lead to a dominant suppression of visually-evoked responses. In tasks where spatiotemporal features (e.g., orientation and direction) are encoded by neuronal populations with bell-shaped, unimodal tuning curves (e.g., over the orientation axis), the interplay between local inhibitory and disinhibitory circuits activated by sound could instead lead to a decrease of perceptual uncertainty and an improvement of decoding accuracy.

Another possible explanation at the root of the discrepancy between our findings and those of some previous studies is that sound may enhance the early portion of a visually evoked response, while suppressing its sustained epoch, as reported by [35]. Given the nature of our task, which required perceptual integration of TF evidence over a prolonged time window, it is possible that inhibition played a dominant role in modulating visual perception. In other tasks, where evidence about the relevant stimulus feature (e.g., orientation) is available within a few tens of milliseconds after stimulus onset [44], enhancement of early visual cortical responses by concomitant sounds may lead to an increased neuronal (and, possibly, perceptual) discriminability of the visual stimuli.

In conclusion, our experiments demonstrate that task-irrelevant auditory signals have a strong influence on the way task-relevant visual information is processed in the rodent brain. By combining psychophysics and computational modeling, our study points to very specific processes underlying these hetero-modal influences. As graphically illustrated in the cartoon of Fig 6, our prediction is that sound bursts suppress visual cortical responses to drifting gratings via cortico-cortical projections from A1 to V1, lowering the gain of the tuning of V1 neurons for the gratings' temporal frequency. This provides clear mechanistic hypotheses that future neurophysiological and neuroanatomical work on visual cortical codes will have to test, in order to fully understand the nature of this modulation and its interplay with task requirements.

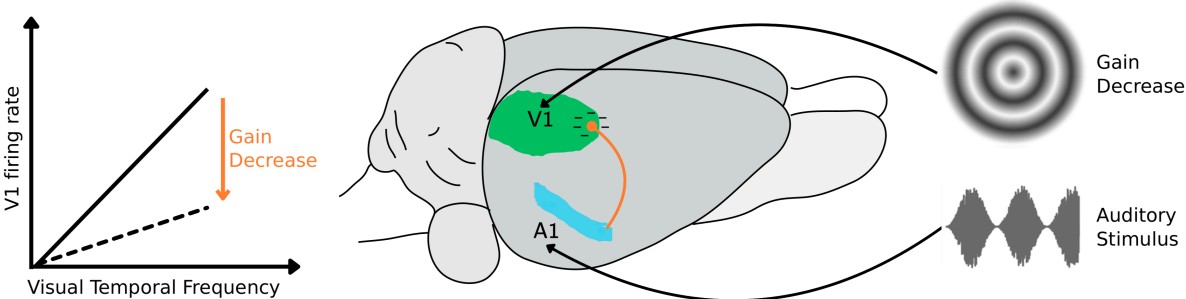

**Fig 6. Outline of the hypothesized interaction between auditory and visual cortex underlying the sound-driven compression of rat visual perceptual space.** Activation of A1 (in cyan) by sound bursts leads to an inhibition of the responses evoked by circular drifting gratings in V1 (in green) via cortico-cortical projections (orange line). This produces a decrease in the gain of the tuning curves of V1 neurons over the temporal frequencies of the gratings.

## Materials and methods

### Subjects

All animal procedures followed international and institutional standards for animal care and use in research and were approved by the Italian Ministry of Health (project approval 940/2015-PR on September 4, 2015; project approval 1183/2020-PR on December 4, 2020).

We trained 12 adult Long Evans rats (Charles River Laboratories) in a visual temporal frequency (TF) classification task. The animals, seven weeks old on arrival, weighed approximately 240 g each. They were housed three per cage in a climate-controlled chamber (23 ± 1° C, 40-50% humidity). Training began at 10 weeks, following two weeks of handling to accustom them to human interaction. Rats always had free access to food (20 g/day) but their access to water was restricted in the days of the behavioral training/test (5 days a week). Water was provided as a reward for correct responses in the discrimination task and for about 30 minutes after each session, ensuring at least the recommended 50 ml kg$^{-1}$ daily intake. During the course of the study, their body weight increased to about 500 g. The duration of the training phase (see below) varied depending on the animal. Five rats completed it in about 3 months and proceeded to the test phase (see below), while for the remaining rats it took about 6 months. The test phase lasted approximately 12 months for all the rats.

### Behavioral apparatus

Rats were trained in a rig made of two identical racks, each with three vertically stacked slots, allowing six animals to be trained simultaneously. Each slot contained a custom-built black operant chamber (20×20×30 cm) with a 4 cm-diameter viewing hole for the rats to extend their heads and face the stimulus display (Fig 1A). The visual stimuli were presented on a 21.5" LCD monitor (ASUS VE228) placed at 30 cm from the viewing hole, while the auditory stimuli were presented from a full-range speaker (Visaton BF37, 4 Ohm, 1.5" diameter, frequency response 100–20000 Hz) positioned above the monitor, tilted 45° towards the viewing hole, and located 35 cm from it. Each speaker received a pre-amplified signal (Q-BAIHE TDA7377PRO). A 3D-printed block with three equidistant response ports was positioned at 3 cm from the viewing hole. Each port had a stainless-steel feeding needle and a LED-photodiode pair that worked as a licking sensor to detect interactions of the rats with the port. Signals from the photodiodes were digitized with a microcontroller (Phidgets 1203 input/output device) and sent to a computer running the open-source MWorks software (https://mworks.github.io/). This system automatically presented the visual and auditory stimuli when the animal's nose was detected in the central port, while the lateral ports were used to record the responses of the animals to the visual stimuli and decide whether to deliver the reward, along with reinforcement sounds (Fig 1B). Correct responses were rewarded with a 4% stevia-water solution from the lateral feeding needles connected to computer-controlled syringe pumps (New Era Pump Systems; NE-500). Each animal was individually monitored with a camera (AXIS M1033-W Network Camera) positioned 14 cm above the chamber viewing hole. The operant chambers were acoustically insulated from each other and the environment with sound-proof foam panels.

### Visual and auditory stimuli

Rats were presented with grayscale outward-moving sinusoidal gratings (30 fps) with a spatial frequency of 0.05 cycles deg$^{-1}$ and temporal frequencies of 0.25, 0.82, 1.32, 1.75, 2.12, 2.48, 2.91, 3.41, and 4 Hz (the temporal frequency of the gratings can be defined as the number of

grating cycles moving through a point of the screen in 1 second). These frequencies were symmetrically distributed around 2.12 Hz on a logarithmic scale (Fig 1C, x axis). The gratings, displayed on a middle-gray background, were enveloped with a Gaussian circular aperture with diameter corresponding to the monitor height (see examples in Fig 1A and 1B).

Two classes of auditory stimuli were used in our study: a fixed amplitude sound (Fig 1D, green curve) and amplitude-modulated sounds (see Fig 1D, gray curve, for an example; see S2A Fig, gray curves, for the complete set of stimuli). In both cases, the stimuli were generated in MATLAB (https://www.mathworks.com/) from a full-range white noise sampled at 44.1 kHz. In the case of the AM sounds, the white noise time series was used as a carrier and multiplied with positive sinusoid envelopes (the modulators), having the same temporal frequencies of the visual stimuli (Fig 1C, y axis). More in detail, sound stimuli were generated as digital control signals, sampled at 44.1 KHz in arbitrary units, which were then converted in WAV format (16 bits per sample). The WAV files were then passed through our custom-built amplifier and loudspeaker (see "Behavioral Apparatus" and Fig 1A).

The control signal was generated starting from a white noise sequence. We used two distinct white noise sequences, one for the fixed amplitude stimulus and one for the AM stimuli. In the fixed amplitude condition, the white noise sequence (denoted as $a_{\text{fix}}(t)$) was composed of independent and identically distributed random variables sampled from a normal distribution with a sampling rate matching that of the control signal ($\nu_{\text{WN}}$ = 44.1 kHz). The distribution had parameters $\mu$ = 0 and $\sigma$ = 1. Before conversion to WAV format the sequence was normalized to have values in the interval [−1,1], hence giving the distribution an effective standard deviation $\sigma_{eff}$ = 0.283. The same sequence was used in all the "V + fixed amplitude sound" trials.

The white noise carrier of the amplitude-modulated sounds, denoted as $a_{AM}(t)$, was composed of independent and identically distributed random variables sampled from a uniform distribution in an interval [−1,1], again with a sampling rate of $\nu_{\text{WN}}$ = 44.1 kHz. The same sequence was used in all the trials of the modulated conditions. The sequence $a_{AM}(t)$ was multiplied by a modulation function $m(t) = (1 - \cos(\omega_n t))/2$, where $\omega_n = 2\pi\nu_n$ is the angular frequency of modulation and $\nu_n$ is the modulation frequency of condition $n$ (S2A Fig, gray curves). The modulation was non-negative and bounded in [0,1], ensuring that the modulated signal remained within [−1,1].

## Estimation of sound intensity

We measured the intensity of the sound stimuli with a phonometer (Tadeto SL720). At any point in time, the phonometer provides an instantaneous estimate of the sound intensity based on a characteristic sampling window (0.125 s). To avoid any interactions between the temporal modulation of our sound stimuli and the finite sampling rate of the phonometer, we devised and calibrated a transformation that estimated the instantaneous sound intensity as a function of the nominal control signal. This transformation takes into account that the control signal $a(t)$ is converted into a sound pressure wave $p(t)$ by the amplifier/speaker, assuming minimal distortion (i.e., that there is a linear relationship between $a$ and $p$), and that the measurements of sound intensities $L(t)$ returned by the phonometer are in dB - i.e., they are a nonlinear function of $p(t)$. We note here that our dB measurements done using "A" frequency weighting, as in previous studies [29].

To carry out this computation, we first verified that the relationship between the nominal control signal $a(t)$ and $p(t)$ as measured by the phonometer was linear. To do so, we measured with the phonometer the intensity $L$ of a uniform white noise sequence lasting 10 seconds. The long duration ensured that we got a precise estimate of the intensity in dB. We

repeated the measurements 10 times, each time dimming the control signal $a$ by a factor of 0.8. All these measures were taken by placing the phonometer inside the sound-proof operant chambers, in the same position where the head of the rats was located during the task. Under the same settings, we also measured the intensity $L_a$ of the ambient noise inside the operant chambers (32 dB). The relationship between the sound intensity $L$ (in decibels) of a white noise sequence and its sound pressure level $p$ (in Pascals) is given by:

$$L = 10 \times \log_{10} \left( \frac{\bar{p}^2 + p_a^2}{p_0^2} \right) \tag{2}$$

where $\bar{p}$ is the root-mean-square (RMS) of $p(t)$ over a time window $\tau$, $p_0 = 20 \ \mu P$ is the reference pressure, and $p_a$ is the pressure of the ambient noise. Inverting the equation, we can compute the sound pressure from the intensity measurements. The ambient sound pressure is given by:

$$p_a = p_0 \sqrt{10^{\frac{L_a}{10}}} \tag{3}$$

The sound pressure of the white noise sequences is, therefore:

$$p = p_0 \sqrt{10^{\frac{L}{10}} - \left( \frac{p_a}{p_0} \right)^2} \tag{4}$$

We could thus compute $p$ from the measured values of $L$, finding that, as expected under the assumption of a linear mapping from $a$ to $p$, the average pressure for each sequence was $0.8 \pm 0.01$ times as intense as the previous one. This confirmed that $p(t) = f \times a(t)$ for some value of $f$ and allowed us to estimate the factor $f$ of the linear relationship as $f = 0.39$ P. We repeated the same procedure with gaussian white noise sequences, confirming the result and obtaining a compatible estimate for $f$. We could then use the estimated value of $f$ to calculate the sound intensity (in dB) of all sound stimuli used in the experiment. For the fixed amplitude, white noise sound $a_{fix}(t)$, this results in $L_{fix} = 74.8$ dB. For the amplitude-modulated stimuli, their sound intensity profile $L_{AM}(t)$ is in principle influenced by the duration $\tau$ of the time window over which the RMS of $p_{AM}(t)$ is computed. For short values of $\tau$, close to the white-noise-sampling period ($T_{WN} = \frac{1}{\nu_{WN}}$), noise fluctuations dominate the temporal profile. By contrast, long $\tau$s, comparable to the period of the modulation, will effectively smooth the modulation, resulting in $L_{AM}$ values that do not accurately represent the dynamic characteristics of the signal. For a wide range of intermediate time windows, however, the computed sound intensity is independent of the details of the fluctuation of the carrier and its temporal profile correctly tracks the temporal modulation.

More specifically, the RMS of $p_{AM}(t)$ over a time window $\tau$ is defined as

$$\bar{p}_{AM}(t)_\tau = \sqrt{\langle f^2 \cdot a_{AM}(t)^2 \cdot m(t)^2 \rangle_\tau} \tag{5}$$

If $\tau$ is smaller than the temporal scale of the modulation, then $m(t)$ is approximately constant inside the integration window and

$$\langle f^2 \cdot a_{AM}(t)^2 \cdot m(t)^2 \rangle_\tau \simeq f^2 m(t)^2 \langle a_{AM}(t)^2 \rangle_\tau \tag{6}$$

On the other hand, if $\tau$ is larger than the sampling rate of the white noise ($\tau \gg T_{WN}$) there will be enough samples in the window for the RMS of $a_{AM}(t)$ to be equal to the uniform

distribution average:

$$\bar{a}_{\text{AM}}^2 = \left\langle a_{\text{AM}}(t)^2 \right\rangle_\tau \simeq \mathbb{E}\left[ a_{\text{AM}}(t)^2 \right]_{p(a_{\text{AM}})} = \int_{-1}^{1} \frac{1}{2} a^2 \, da = \frac{1}{3} \tag{7}$$

Therefore, the resulting temporal intensity profile of the AM sounds (in dB), when using one of the intermediate values of $\tau$ can be computed as:

$$L_{\text{AM}}(t) = 10 \log_{10} \left( \left( \frac{f \cdot \bar{a}_{\text{AM}} \left( \frac{1 - \cos(\omega t)}{2} \right)}{p_0} \right)^2 + \left( \frac{p_a}{p_0} \right)^2 \right) \tag{8}$$

The sound intensity profiles $L_{\text{AM}}(t)$ for each modulation frequency (computed with $\tau = 5$ ms $\simeq 220 T_{\text{WN}}$), as well as the profiles of $L_{\text{fix}}$ and $L_a$ (with the latter corresponding to the "visual only" condition) are shown in S2B Fig.

To estimate the sound intensity experienced by a rat during each trial, we computed the average intensity $\bar{L}_n(T)$ of the trial condition $n$ throughout a specific time window with duration $T$. This duration was equal to the *reaction time* ($RcT$) of the animal in that given trial, i.e., to the period during which the rat was exposed to the sound stimulus before initiating a motor response. The reaction time is defined as:

$$RcT = RT - MoRsT \tag{9}$$

where $RT$ is the *response time* (i.e., the time from the trial onset to the moment a response port is reached), and $MoRsT$ is the *motor response time* (i.e., the time taken by the animal to reach the selected response port, after leaving the central port for the first time). In the recorded data, only the response time was directly measured for each trial. To estimate the reaction time, we assigned to $MoRsT$ a fixed value across all trials, as determined in one of our previous studies, which carefully mapped the trajectories and durations of rat head movements in a similar experimental setup [53]. Specifically, we used $MoRsT = 0.3$ s, which corresponds to the duration of a direct ballistic movement from the central port to either response port. This type of movement was the most frequently observed in that study (1149 out of 1359 recordings). Hence, for each trial $i$, we computed the average perceived intensity as $\bar{L}_n(RcT_i) = \bar{L}_n(RT_i - MoRsT)$. An example of the computation is shown on the "0.25 Hz AM sound" row of S2B Fig.

The resulting values for each trial are shown as scatter plots in S2C Fig, along with violin plots to illustrate their distribution in each sound condition for every individual rat (the dashed lines report the rat group average intensities). The averages of these sound intensities obtained for each rat are those reported on the x axis of Fig 5B (squares, triangles, and crosses). Note, however, that, in that figure, the AM conditions with modulation frequencies 1.32 Hz $\leq$ TF $\leq$ 4 Hz have been grouped together, given that their average intensity was virtually identical (as shown in Fig 1D).

## Behavioral task

The rats learned to initiate a behavioral trial by nose-poking the central port, and to approach the lateral ports to classify the visual stimuli based on their temporal frequency (Fig 1B). The "low temporal frequency" class (0.25-1.75 Hz) and the "high temporal frequency" class (2.48-4 Hz) had to be reported by half of the rats by poking, respectively, the left and right port, while the other half was trained with the opposite association. To initiate a trial, rats

had to keep their nose in the central port for 300 ms, followed by a 100 ms tone (500 Hz pure tone) signaling the stimulus presentation, occurring 200 ms later. After stimulus onset, rats were given 2 seconds to make a response; responses within 300 ms were aborted to prevent impulsivity. Correct responses were rewarded with 4% stevia-water solution and a concomitant reinforcement sound, while incorrect responses were followed by an unpleasant sound (400 ms, 300 Hz square-wave) and a 1-3 second timeout period. A middle-grey background was displayed during non-stimulus periods. In a given trial, every stimulus had an equal probability of being presented, but with the constraint that a maximum of three consecutive presentations of the same visual class was allowed to prevent the development of a bias towards one of the response ports. This trial structure was the same for all the training and test phases of the experiment, described in the next section. Stimulus presentation, response collection, and reward delivery were controlled by the MWORKS software (https://mworks.github.io).

## Experimental design

**Training.** The animals were trained according to the following phases.

*Habituation*: the rats were first habituated to the operant chamber and were gradually trained to initiate trials autonomously and receive rewards from the lateral feeding needles. Initially, the animals were manually guided to the nose-poke ports, often reinforced with honey drops. A trial started once the animal touched the central port with its nose, signaled by a positive reinforcement tone, and followed by stevia-water rewards from both lateral feeding needles. This phase lasted about 7 days, progressing to the next phase once the animals initiated at least 100 trials autonomously in three consecutive sessions.

*Phase 1*: rats were conditioned to associate rewards with the presentation of paired auditory and visual stimuli. Initially, to facilitate the task, the visual stimuli were those with the lowest (0.25 Hz) and highest (4 Hz) temporal frequencies and were presented at full contrast alongside the fixed amplitude white noise sound for 2 seconds. Concomitantly, the reward was delivered from the correct response port, based on the class the visual stimulus belonged to. Advancement to the next phase was based on qualitative inspection of the behavior, ensuring that rats attended to the visual stimuli. Importantly, here, as in the rest of the training and test phases, only the visual stimuli were associated to a given reward port (i.e., to a given response class). The auditory stimuli were always task-irrelevant, i.e., were uninformative about the correct classification of the visual stimuli.

*Phase 2*: in this phase, the rats were actively reinforced based on their classification of the visual stimuli, presented again with a temporal frequency of either 0.25 or 4 Hz. That is, in this phase, the reward was delivered only after the rat had approached and licked the correct response port, while, in case of incorrect choice, the disturbing sound was delivered and the time-out period started. Again, both visual stimuli were paired with the same, uninformative fixed amplitude sound. Advancement to the next phase required that an animal correctly classified at least 70% of the visual stimuli in at least two consecutive sessions. Two of the 12 rats failed to meet this criterion and were excluded from further testing.

*Phases 3, 4, and 5.*: here, we gradually introduced the additional temporal frequencies of the visual stimuli. First, we added the gratings at 0.82 and 3.41 Hz (*Phase 3*), then those at 1.32 and 2.91 Hz (*Phase 4*), bringing to six the total of stimuli to classify. Finally, we presented all nine possible visual temporal frequencies (*Phase 5*). Also in this case, each advancement to the next phase depended on whether an animal was able to achieve an accuracy criterion of 70% correct responses in at least two consecutive training sessions. The 2.12 Hz stimulus, serving as an arbitrary boundary between the "low temporal frequency" and "high temporal

frequency" classes, was randomly assigned to one of the classes in each trial and rewarded accordingly.

Upon achieving 70% accuracy in at least two consecutive sessions of Phase 5 (i.e., once all visual temporal frequencies had been introduced), indicating proficiency in classifying all visual stimuli, we progressively reduced the contrast of the visual stimuli. The contrast reduction was performed in two main blocks based on accuracy criteria. In the initial exploratory block of contrast reduction, stimulus contrast was progressively lowered within 6-10 sessions, bringing it down to 15% for some animals (e.g., see Example Rat 1 in Fig 2A). During this phase, transient drops in accuracy occasionally occurred; in such cases, the contrast was temporarily increased to the preceding level or back to 100% to allow the animal to recover and consolidate its performance before resuming the reduction (e.g., see Example Rat 2 in Fig 2B). To further ensure the robustness of learning and to verify that animals generalized their classification strategy across contrast levels, all animals were subsequently re-exposed to full contrast before undergoing a second, final contrast reduction block. This final phase was implemented in three steps (50%, 35%, and 25% contrast), based on accuracy criteria: if a rat maintained ≥70% correct choices for at least two consecutive sessions at a given contrast level, the contrast was further decreased. The learning trajectory during the training phase is illustrated in Fig 2 for two representative rats. All animals consistently reached stable performance at 25% contrast across temporal frequencies, which defined the entry point to the following test phase.

Each training session lasted 40-60 minutes [39,40], with the number of trials per session varying between ∼100 and ∼400 depending on the rat (with some animals being more prolific than others) and on the specific training day. As shown in Fig 2B, occasional drops in trial counts occurred, typically associated with a decrease in performance, which were likely caused by momentary reductions in the motivation of the animal or temporary failures of the behavioral rig (e.g., uneven delivery of liquid from the reward ports).

**Test phase.**   After completing the training phase, the rats entered the test phase. Here, the visual stimuli presented in each trial could be paired not only with the fixed amplitude sound (as done during training) but also with the amplitude-modulated auditory stimuli. Specifically, the trials in the test phase fell into four main categories, based on the combination of visual and auditory stimuli (see Fig 1C).

- In the "V + fixed amplitude sound" trials, which matched those in training Phase 5, the visual gratings of varying temporal frequencies were paired with the fixed amplitude white noise sound (green cells in Fig 1C).
- In the "V + AM sounds" trials, the visual gratings were paired with each of the amplitude-modulated sounds at the various frequencies, resulting in all 81 possible pairwise combinations of visual and auditory frequencies (cells in the squared matrix of Fig 1C).
- In the "V only" or unimodal visual trials, the drifting gratings were presented without any accompanying sound (purple cells in Fig 1C).
- In the "A only" or unimodal auditory trials, the amplitude-modulated sounds were presented alone, without the visual gratings (cyan cells in Fig 1C). Since the rats had never been trained to classify the auditory stimuli, we simply collected their spontaneous responses to these sounds, without providing any feedback (i.e., neither reward nor time-out) about their choices.

All the stimulus conditions listed above were presented (pseudo-randomly interleaved) in every test session. As during the training phases, each test session lasted 40-60 minutes

[39,40], with the number of trials per session varying between ~100 and ~400 depending on the rat and on the specific training day. The typical number fo test sessions that were collected for each animal was ~90.

Again, throughout both the training and test phases, the temporal frequency of the visual stimuli remained the only task-relevant feature for inferring the correct reward location. In contrast, the auditory stimuli were always task-irrelevant and did not indicate the reward side. We verified that the rats did not implicitly learn to classify the auditory stimuli according to their temporal frequency, by plotting their choices in the "A only" trials (see S1 Fig). The resulting group average psychometric curve was flat, showing no modulation as a function of sound frequency.

## Statistical analyses

The classification accuracies achieved by the rats in the test conditions described in the previous section were statistically compared using a repeated measures, two-way ANOVA (see Fig 3 and Tables 1 and 2). To perform the ANOVA, we first calculated the probabilities of each rat to classify a visual stimulus as belonging to the "High TF" category across all experimental conditions. This resulted in 10 data points for each of the stimulus conditions analyzed in Fig 3A and 3B, corresponding to the 10 rats in our study. The ANOVA analysis requires two key assumptions for the data distributions:

1. Normality, i.e., the data should follow a Gaussian distribution. This is particularly important for proportions of correct choices, as distributions near 0 and 1 can saturate and deviate from normality. To address this issue, we transformed the psychometric data using the arcsine of the square root of each value. After the transformation, we tested the normality of the distributions with the Shapiro-Wilk test [54] and we verified that it was satisfied by each data point distribution.

2. Homoscedasticity, i.e., the variance should be equal across experimental conditions. The arcsine transform also helps stabilize the variance for proportional data. This was assessed using Levene's test. Following the transform, the test could not reject the null hypothesis that all distributions had equal variance ($p = 0.3421$), indicating that the assumption of homoscedasticity was satisfied.

We then proceeded to calculate the two-way ANOVA with repeated measures. Our experimental design involved two independent variables: the TF of the visual stimuli, with 9 levels, and the experimental condition, with 3 levels. The latter were as follows. For the analysis carried out in Fig 3A: the "V + fixed amplitude sound", the "V + AM sounds (congruent)" and the "V + AM sounds (anti-congruent)"; for the analysis carried out in Fig 3B: the "V + fixed amplitude sound", the "V + AM sounds" and the "V only". The dependent variable was rat response, measured as the probability of classifying a visual stimulus in the "High TF" category for each experimental condition, which was recorded for the same 10 animals across all the different levels of the independent variables. Tables 1A and 2A present the results of this analysis for the data shown, respectively, in Fig 3A and 3B, reporting, in both cases, significant effects for both the visual TF and the experimental condition variables, as well as a significant interaction between them. This indicates that the dependent variable changed across different visual TFs and experimental conditions, and that there was a combined influence of visual TF and experimental condition on the dependent variable.

Finally, to determine which specific levels of the experimental conditions contributed to the main effect reported in the ANOVA table, we conducted post-hoc tests. We performed

pairwise comparisons of the different levels of the experimental conditions and evaluated their effects using the Tukey test to control for multiple comparisons, as summarized in Tables 1B and 2B.

## Ideal observer model

In this section we derive the ideal observer model used to predict the responses of the rats in the visual rate classification task. The ideal observer follows a standard structure for a sensory classification task, based on the assumption that an internal, abstract representation of the visual stimulus, called the *measurement* and indicated by *x*, can be modeled as a Gaussian random variable (an approach that is firmly established in psychophysical modeling, going back to its roots in signal detection theory, see e.g. [46,47,55–57]). The main nonstandard element of our analysis is the added hypothesis that *x* is scaled by a gain factor $\gamma$ which can depend on the intensity of the auditory stimulus. This additional hypothesis stands on its own on an abstract level, but we show below that it can be derived as the consequence of a putative suppressive effect of sound on the activity of visual cortical neurons.

On each trial of the two-alternative forced choice (2AFC) task, the visual stimuli were assigned a temporal frequency which was randomly chosen from a pool of 9 possible values (0.25, 0.82, 1.32, 1.75, 2.12, 2.48, 2.91, 3.41 and 4 Hz). In the following, we will denote this visual temporal frequency by *s*, for stimulus. The goal of the animal was to accurately classify each visual stimulus as belonging to the low frequency class (*L*, for gratings with frequencies from 0.25 to 2.12 Hz) or to the high frequency class (*H*, from 2.12 to 4 Hz). Each of the 9 frequency values had an equal chance of being presented in a trial. The central value was randomly assigned to the *H* or *L* class, and therefore also *H* and *L* had an equal chance of being presented. The probabilistic structure of the task can also be restated as follows. On each trial, the stimulus class *C* is chosen with equal chance to be *H* or *L*, that is

$$p(C = H) = p(C = L) = \frac{1}{2} \tag{10}$$

Based on the class, the visual stimulus *s* presented to the animal is selected with probability

$$p(s|C) = \frac{\delta(s - s_0)}{2K + 1} + \frac{2}{2K + 1} \sum_{k=1}^{K} \delta(s - s_k^C) \tag{11}$$

where

$$s_0 = 2.12 \tag{12}$$
$$s_k^L \in \{0.25; 0.82; 1.32; 1.75\} \tag{13}$$
$$s_k^H \in \{2.48; 2.91; 3.41; 4\} \tag{14}$$
$$K = 4 \tag{15}$$

and for convenience with our derivations below we order the index *k* symmetrically around $s_0$, such that $s_1^H = 2.48$, $s_2^H = 2.91$, etc and $s_1^L = 1.75$, $s_2^L = 1.32$, etc. Our task here is to compute the probability $P(\hat{C} = H \mid s, n)$ that the observer chooses the class "high visual frequency" ($\hat{C} = H$), when the frequency of the visual stimulus is *s* and the sound condition is *n*.

We start by modeling the process that on each trial gives rise to the measurement *x*, given the visual stimulus *s* and the sound condition *n*. We initially disregard the effect of the sound, which we will add back later. We assume that a stimulus *s* gives rise to a certain pattern $\vec{r}$ of

firing rates in V1, where $r_i$ is the rate of the $i$-th neuron. The population response is normally distributed around a stimulus-dependent value $\vec{\mu}(s)$: $\vec{r} \sim \mathcal{N}(\vec{\mu}(s), \Sigma)$, where $\Sigma$ is a generic (but stimulus-independent) covariance matrix.

The subject generates an internal representation (the *measurement*) $x$ of the external stimulus $s$ based on the activation pattern $\vec{r}$. We assume that this measurement is a projection of the vector of activities $\vec{r}$ on some measurement axis $y = \vec{u} + t\vec{v}$; in other words,

$$x = (\vec{r} - \vec{u}) \cdot \vec{v} \tag{16}$$

This is biologically plausible if, for example, the measurement $x$ is the activity of a readout neuron with connectivity strength $v_i$ with each neuron coding $s$. Ideally, the measurement axis $y$ used by the subject is the one that allows to best discriminate the stimulus classes, therefore allowing for the best performance, but for the purpose of developing the model we do not need to make any particular assumptions on $\vec{u}$ and $\vec{v}$. Because $x$ is an affine transformation of a normally-distributed random variable, it is itself a normally distributed random variable:

$$x \sim \mathcal{N}\left((\vec{\mu}(s) - \vec{u}) \cdot \vec{v}, \vec{v}^\mathsf{T}\Sigma\vec{v}\right) \tag{17}$$

If the projection operation is itself noisy and introduces additional Gaussian noise of standard deviation $\tau$, the distribution of the measurement becomes simply

$$x \sim \mathcal{N}\left((\vec{\mu}(s) - \vec{u}) \cdot \vec{v}, \vec{v}^\mathsf{T}\Sigma\vec{v} + \tau^2\right) \tag{18}$$

In the following, we will lump both sources of variability into a single parameter $\rho$,

$$\rho^2 = \vec{v}^\mathsf{T}\Sigma\vec{v} + \tau^2 \tag{19}$$

We will conceptualize $\rho$ as the general level of sensory noise present in the system for the purposes of this task. The distribution of the measurement will therefore be

$$x \sim \mathcal{N}\left((\vec{\mu}(s) - \vec{u}) \cdot \vec{v}, \rho^2\right) = \mathcal{N}\left(\nu(s), \rho^2\right) \tag{20}$$

where we have defined

$$\nu(s) = (\vec{\mu}(s) - \vec{u}) \cdot \vec{v} \tag{21}$$

for notational convenience.

To decide if a stimulus has high frequency ($H$) or low frequency ($L$), we assume that the subject performs Bayesian inference to compute a posterior probability over the classes $H$ and $L$ given the measurement and reports the class with highest posterior. In our case (see below for a detailed justification), this will be equivalent to finding a decision boundary $x^*$ such that $P(H \mid x^*) = P(L \mid x^*)$, and reporting $H$ if $x > x^*$ and $L$ if $x < x^*$. On average over the distribution of the measurement given the stimulus, the probability of reporting a given class given the stimulus (which we may also call the psychometric function) will then be

$$p(\hat{C} = H \mid s) = \int_{x:p(H|x)>p(L|x)} p(x \mid s)dx \tag{22}$$

$$= p(x > x^* \mid s) \tag{23}$$

$$= \int_{x^*}^{\infty} p(x \mid s)dx \tag{24}$$

and $p(\hat{C} = L \mid s) = 1 - P(\hat{C} = H \mid s)$. We will now calculate $x^*$ and consequently we will derive $p(\hat{C} = H|s)$.

Let's consider the simplest possible scenario for $\vec{\mu}$, namely the case where $\vec{\mu}$ is linear in $s$:

$$\vec{\mu}(s) = \vec{\alpha} + \vec{\beta}s \tag{25}$$

In this case, the mapping between $s$ and $x$ also becomes linear:

$$\nu(s) = (\vec{\mu}(s) - \vec{u}) \cdot \vec{v} = \vec{\alpha} \cdot \vec{v} + (\vec{\beta} \cdot \vec{v})s - \vec{u} \cdot \vec{v} = a + bs \tag{26}$$

with $a = (\vec{\alpha} - \vec{u}) \cdot \vec{v}$ and $b = \vec{\beta} \cdot \vec{v}$.

We will now show that when $\vec{\mu}$ is linear the symmetry of the problem and the shape of the response distributions imply that $x^*$ is unique and $x^* = \nu(s_0)$.

If we compute the posteriors explicitly from the class-conditional stimulus distributions in Eq (11) we get

$$p(H|x) - p(L|x) = \frac{p(x|H)p(H)}{p(x)} - \frac{p(x|L)p(L)}{p(x)} \tag{27}$$

$$\propto p(x \mid H) - p(x \mid L) \tag{28}$$

$$= \frac{p(x|s_0)}{2K+1} + \frac{2}{2K+1} \sum_{k=1}^{K} p(x|s_k^H) - \frac{p(x|s_0)}{2K+1} - \frac{2}{2K+1} \sum_{k=1}^{K} p(x|s_k^L) \tag{29}$$

$$\propto \sum_{k=1}^{K} p(x|s_k^H) - p(x|s_k^L) \tag{30}$$

$$\propto \sum_{k=1}^{K} \exp\left[-\frac{(x - \nu(s_k^H))^2}{2\rho^2}\right] - \exp\left[-\frac{(x - \nu(s_k^L))^2}{2\rho^2}\right] \tag{31}$$

$$= \sum_{k=1}^{K} \exp\left[-\frac{(x - a - bs_k^H)^2}{2\rho^2}\right] - \exp\left[-\frac{(x - a - bs_k^L)^2}{2\rho^2}\right] \tag{32}$$

Note now that the symmetry of the class-conditional stimulus distributions in Eq (11) is such that

$$\begin{cases} (s_0 - s_k^H)^2 = (s_0 - s_k^L)^2 \; \forall k \\ (s - s_k^H)^2 < (s - s_k^L)^2 \; \forall k & \text{if } s > s_0 \\ (s - s_k^H)^2 > (s - s_k^L)^2 \; \forall k & \text{if } s < s_0 \end{cases} \tag{33}$$

Also, for any value of $x$ and $s$,

$$x - \nu(s) = a - b\nu^{-1}(x) - a - bs = b(\nu^{-1}(x) - s) \tag{34}$$

which means that, if $b > 0$ (and simply reversing the inequalities if $b < 0$),

$$\begin{cases} (x - \nu(s_k^H))^2 = (x - \nu(s_k^L))^2 \; \forall k \\ (x - \nu(s_k^H))^2 < (x - \nu(s_k^L))^2 \; \forall k & \text{if } x > \nu(s_0) \\ (x - \nu(s_k^H))^2 > (x - \nu(s_k^L))^2 \; \forall k & \text{if } x < \nu(s_0) \end{cases} \tag{35}$$

Therefore,

$$\begin{cases} p(H|x) = p(L|x) & \text{if } x = \nu(s_0) = a + b s_0 \\ p(H|x) > p(L|x) & \text{if } x > \nu(s_0) = a + b s_0 \\ p(H|x) < p(L|x) & \text{if } x < \nu(s_0) = a + b s_0 \end{cases} \tag{36}$$

Thus as a consequence of the symmetry of the problem the decision boundary $x^*$ is simply

$$x^* = \nu(s_0) = a + b s_0 = (\vec{\alpha} + \vec{\beta} s_0 - \vec{u}) \cdot \vec{v} \tag{37}$$

Hence:

$$p(\hat{C} = H \,|\, s) = p(x > x^* \,|\, s) = \int_{x^*}^{\infty} p(x \,|\, s) dx \tag{38}$$

$$= \int_{x^*}^{\infty} \frac{1}{\sqrt{2\pi\rho^2}} \exp\left[ -\frac{(x - (\vec{\mu}(s) - \vec{u}) \cdot \vec{v})^2}{2\rho^2} \right] dx \tag{39}$$

$$= \Phi\left[ \frac{(\vec{\mu}(s) - \vec{u}) \cdot \vec{v} - x^*}{\rho} \right] \tag{40}$$

$$= \Phi\left[ \frac{(\vec{\mu}(s) - \vec{u}) \cdot \vec{v} - (\vec{\mu}(s_0) - \vec{u}) \cdot \vec{v}}{\rho} \right] \tag{41}$$

$$= \Phi\left[ \frac{(\vec{\mu}(s) - \vec{\mu}(s_0)) \cdot \vec{v}}{\rho} \right] \tag{42}$$

$$= \Phi\left[ \frac{\vec{\beta}(s - s_0) \cdot \vec{v}}{\rho} \right] \tag{43}$$

$$= \Phi\left[ \frac{s - s_0}{\sigma} \right] \tag{44}$$

with

$$\sigma = \frac{\rho}{\vec{\beta} \cdot \vec{v}} \tag{45}$$

As shown in Fig 4, we model the effect of an external sound stimulus $n$ as a suppression of the activity of V1 neurons. This suppression may result from direct inhibition of V1 neurons by auditory neurons, but it could arise from indirect effects as well. We model it by multiplying the mean response $\vec{\mu}(s)$ of the neurons by a sound-dependent factor $\gamma_n$, while keeping fixed the decision boundary $x^*$ learned during the training sessions. By substituting $\vec{\mu}(s)$ with $\gamma_n \vec{\mu}(s)$ in the integral above, we get:

$$p(\hat{C} = H \,|\, s, n) = \Phi\left[ \frac{(\gamma_n \vec{\alpha} + \gamma_n \vec{\beta} s - \vec{u}) \cdot \vec{v} - x^*}{\rho} \right] = \Phi\left[ \frac{\gamma_n s - s_0 + (\gamma_n - 1)\lambda}{\sigma} \right] \tag{46}$$

with

$$\lambda = \frac{\vec{\alpha} \cdot \vec{v}}{\vec{\beta} \cdot \vec{v}} \tag{47}$$

Now, if the circuit computing $x$ is adapted to extract the maximum amount of information from $\vec{r}$ in this task, at least in the case where the noise for $\vec{r}$ is isotropic, $\vec{v}$ will be parallel to $\vec{\beta}$ (this may not be true in presence of nontrivial noise correlation structure). On the other hand, if $\vec{\alpha}$ is a random high dimensional vector with no particular relation to $\vec{v}$, we can assume that with high probability $\vec{\alpha} \cdot \vec{v} = 0$. Therefore, under these simple assumptions, $\lambda = \vec{\alpha} \cdot \vec{v}/\vec{\beta} \cdot \vec{v} = 0$. In this case, the expression for the probability of reporting $H$ in presence of auditory stimulus $n$ reduces to

$$p\left(\hat{C} = H \mid s, n\right) = \Phi\left[\frac{\gamma_n s - s_0}{\sigma}\right] \tag{48}$$

where $\gamma_n$ and $\sigma$ are free parameters of the model. Finally, we note that while the linearity assumption for $\vec{\mu}(s)$ made in this section can seem restrictive, the derivations above remain valid as long as $\vec{\mu}(s) \cdot \vec{v}$ is (approximately) linear over the range of values of $s$ that are relevant in the task.

**Lapse rates.** We extended the ideal observer to account for lapses by defining two lapse rates $\epsilon_H$ and $\epsilon_L$, such that on each trial the observer has a probability $1 - \epsilon_H - \epsilon_L$ of actually engaging in the task, a probability $\epsilon_H$ of reporting $H$ independently of the stimulus, and a probability $\epsilon_L$ of reporting $L$ independently of the stimulus. Therefore,

$$p(\hat{C} = H \mid s, n) = \epsilon_H + (1 - \epsilon_H - \epsilon_L)\,\Phi\left[\frac{\gamma_n s - s_0}{\sigma}\right] \tag{49}$$

and

$$p(\hat{C} = L \mid s, n) = 1 - p(\hat{C} = H \mid s, n) \tag{50}$$

## Inferring the model parameters from experimental data

We used the ideal observer model derived above to describe the data from our experiment. According to the formula for $p(\hat{C} = H \mid s, n)$, the behavior of each rat $r$ is characterized by some parameters $\theta^r = \{\sigma^r, \{\gamma_n^r\}, \epsilon_L^r, \epsilon_H^r\}$. We inferred the value of these parameters using a hierarchical Bayesian approach [58,59], for which we now give a brief conceptual overview (refer to S3 Fig for a graphical depiction).

Denote by $D$ the set of recorded behavioral data, with $D_t^r$ being the choice ($H$ or $L$) of rat number $r \in \{1, \dots, R\}$ on trial number $t \in \{1, \dots, T_r\}$. We start by defining, for any rat, a prior probability for the parameters:

$$p(\theta^r) = p(\sigma^r)p(\epsilon_L^r, \epsilon_H^r)\prod_n p(\gamma_n^r) \tag{51}$$

We also define a likelihood function for the parameters $\theta^r$. The likelihood is simply the probability of observing the rat choices $D^r = \{D_t^r\}_{t=1}^{T_r}$, given a certain value of $\theta^r$ and of the visual and auditory stimuli provided to the rat on all trials (respectively, $s^r = \{s_t^r\}_{t=1}^{T_r}$ and $n^r = \{n_t^r\}_{t=1}^{T_r}$), seen as a function of $\theta^r$:

$$\mathcal{L}^r(\theta^r) = p(D^r \mid \theta^r; s^r, n^r) \tag{52}$$

$$= \prod_{t=1}^{T_r} p\left(\hat{C} = D_t^r \mid \theta^r; s_t^r, n_t^r\right) \tag{53}$$

$$= \prod_{t:D_t^r=H} p\left(\hat{C} = H \mid \theta^r; s_t^r, n_t^r\right) \prod_{t:D_t^r=L} p\left(\hat{C} = L \mid \theta^r; s_t^r, n_t^r\right) \tag{54}$$

where $p(\hat{C} = H|s, n)$ and $p(\hat{C} = L|s, n)$ are given by Eq (49) and Eq (50). The likelihood function therefore encapsulates the ideal observer model, developed in the previous section. By applying Bayes' theorem, we compute a joint posterior probability distribution for the parameters:

$$p(\theta^r \mid D^r) \propto p(D^r \mid \theta^r) p(\theta^r) \tag{55}$$

(from now on, for convenience, we will omit the explicit dependency on $s$ and $n$ in the likelihood). However, since the rats belong to the same population and are subjected to the same experimental conditions, we assume that the parameters describing different animals will be related. In other words, gathering information about one rat tells us something about that specific rat, but should also inform us on what to expect about other rats, or rat behavior in general. We can build this assumption into the model by imposing that the parameters associated to different rats are all sampled from the same distribution, leaving the parameters of this higher-level probability distribution to be also inferred from the data. Besides extracting "population-level" information from the data of each rat, this procedure will also help constraining the rat-level parameters to reasonable values given what we observed in the other rats, thus acting as a regularizer for the inference process. More formally, our posterior will be:

$$p(\theta, \eta \mid D) \propto p(D \mid \theta) p(\theta \mid \eta) p(\eta) \tag{56}$$

where $\theta = \{\theta^r\}_{r=1}^R$ are the rat specific parameters for all $R$ rats, and $\eta$ are the common (population) parameters.

More concretely, the priors we assign to the parameters ($p(\theta \mid \eta)$ in Eq (56)) are:

$$(\epsilon_L^r, \epsilon_H^r, 1 - \epsilon_L^r - \epsilon_H^r) \sim \text{Dirichlet}(1, 1, 1) \quad \forall r \tag{57}$$

$$\sigma^r \sim \text{Gamma}(k_\sigma, \theta_\sigma) \quad \forall r \tag{58}$$

$$\gamma_n^r \sim \mathcal{N}(\mu_{\gamma_n}, \sigma_{\gamma_n}^2) \ \forall r, n \tag{59}$$

It follows that our population parameters $\eta$ are the mean $\mu_{\gamma_n}$ and the standard deviation $\sigma_{\gamma_n}$ of the $\gamma_n$ parameters, and $k_\sigma$ and $\theta_\sigma$ for the perceptual noise. Following standard practice [58,60], we assign weakly informative priors $p(\eta)$ to the population-level parameters:

$$\mu_{\gamma_n} \sim \mathcal{N}(\mu = 0, \sigma^2 = 9) \quad \forall n \tag{60}$$

$$\sigma_{\gamma_n} \sim \text{Exponential}(\text{scale} = 3) \quad \forall n \tag{61}$$

$$k_\sigma \sim \text{Exponential}(3) \tag{62}$$

$$\theta_\sigma \sim \text{Exponential}(3) \tag{63}$$

Finally, in this hierarchical scheme we also need a global likelihood function ($p(D \mid \theta)$ in Eq (56)) which we obtain simply as the product of the rat-level likelihoods:

$$\mathcal{L}(\theta) = p\left(D \mid \{\vec{\gamma}_r, \sigma_r, \vec{\epsilon}_r\}_{r=1}^R\right) \tag{64}$$

$$= \prod_{r=1}^R \prod_{t=1}^{T_r} p\left(\hat{C} = D_t^r \mid \theta^r; s_t^r, n_t^r\right) \tag{65}$$

**Prior predictive check.** We conducted a prior predictive check [58] to evaluate whether the model's prior distributions were consistent with plausible observations, even before any actual data was considered. The check was performed by sampling 1000 sets of model parameters from their prior distributions and using them to generate synthetic data. The results showed that our assumed priors are flexible enough to allow a full range of behavior for the psychometric curves, and have no bias towards supporting the central claim of our paper, namely the dependence of the gain on the sound intensity (S4 Fig).

**Sampling the posterior distribution.** We performed inference with a Markov Chain Monte Carlo (MCMC) method, using the NUTS algorithm [61], as implemented in PyMC version 5.2.0 [62]. This method samples parameters from the posterior distribution $p(\theta \mid D)$: with a high enough number of samples, the distribution of the samples matches closely the distribution of the posterior. We sampled 4 independent chains of 2000 draws each, 1000 of which we used for tuning the algorithm and subsequently discarded, and 1000 as actual samples of the target posterior distribution. The target acceptance probability parameter for NUTS was set at 0.99, and no divergences were detected during sampling.

**Posterior predictive check.** Following the model's inference, we performed a posterior predictive check [58] to evaluate the goodness-of-fit to the observed data. The check was performed by sampling a set of synthetic observations from each of the 1000 parameter posterior draws saved in the Markov chain. The results showed that the simulated data was consistent with the observed data, indicating that the model provides a robust description of the observed rats' behavior (S5 Fig).

## Model comparison

To verify that the derived model describes well the experimental data, we compared it with other models that could also reproduce the psychometric curves of the rats. To perform the comparison, we used a Leave-One-Out Cross-Validation (*LOO*) estimate of the Expected Log pointwise Predictive Density (*ELPD*)[63]. LOO estimates how well, on average, the model describes data it hasn't been trained on. To do so, it computes how well a model trained on all the trials (across all animals) except one predicts the outcome of that trial, and averages this value over all the trials. The ELPD therefore automatically balances goodness of fit with model complexity, penalizing overly complex models that tend to overfit the noise in the data. More formally, the LOO estimate of the ELPD for a model $\mathcal{M}$ for which we can sample the posterior distribution with MCMC is defined as:

$$\mathrm{ELPD_{LOO}}(\mathcal{M}) = \sum_{i=1}^{N} \log \left( \frac{1}{S} \sum_{j=1}^{S} p_{\mathcal{M}}(d_i \mid \theta_{(j;D_{-i})}) \right) \tag{66}$$

where $N = \sum_{r=1}^{R} T_r$ is the total number of trials in the data, $S$ is the number of posterior samples in the Markov chain for model $\mathcal{M}$, $p_{\mathcal{M}}$ is the likelihood function of the model, $d_i$ is the data from the $i$-th trial and $\theta_{(j;D_{-i})}$ is the $j$-th posterior sample of a Markov chain generated from all data except $d_i$ (note that each posterior distribution sample, including $\theta_{(j;D_{-i})}$, is a high-dimensional vector containing one value for each parameter in the model). In Eq (66), the argument of the logarithm represents the likelihood of the model $\mathcal{M}$ for the datapoint $d_i$, averaged over the posterior distribution over $\theta$ (where this posterior is obtained without looking at datapoint $d_i$). Indeed, since the training of $\mathcal{M}$ results in a posterior distribution rather than a single parameter vector, all quantities related to $\mathcal{M}$ must be averaged over this distribution. For MCMC training, this average is computed as a sum over all sampled parameters,

given that the samples follow the posterior distribution:

$$\frac{1}{S}\sum_{j=1}^{S} p_{\mathcal{M}}(d_i \mid \theta_{(j;D_{-i})}) \simeq \int p_{\mathcal{M}}(d_i \mid \theta)p(\theta \mid D_{-i})d\theta = \langle p_{\mathcal{M}}(d_i \mid \theta)\rangle_{p(\theta \mid D_{-i})}$$

In practice, computing Eq (66) requires training the model $N$ times, one per trial: given the large number of trials in our experiment, we approximated the leave-one-out procedure with pareto-smoothed importance sampling (PSIS-LOO) [63].

Computing the ELPD$_{\text{LOO}}$ for different models gives us a relative measure of how well they describe the data. As shown in the inset of Fig 5A, the model with the highest ELPD is the one with 5 distinct $\gamma$ parameters, one for each sound condition whose power is in a certain range (see main text for details).

We also compared this model with a model that incorporates the additional $\lambda$ parameter discussed in the ideal observer derivation above (Eq (47)). This parameter is included with an analogous hierarchical structure to that of $\sigma$ in the main model discussed above, such that each rat has its own $\lambda^r$, with

$$\lambda^r \sim \mathcal{N}(\mu_\lambda, \sigma_\lambda^2) \quad \forall r \tag{67}$$

$$\mu_\lambda \sim \mathcal{N}(\mu = 0, \sigma^2 = 9) \tag{68}$$

$$\sigma_\lambda \sim \text{Exponential}(3) \tag{69}$$

This comparison allowed us to test the hypothesis that, to the extent that the assumptions about stimulus encoding made in that section are valid, the vector of resting activities of neurons $\vec{\alpha}$ has no effect on the projection of measurement. The resulting $(5\gamma 1\lambda 1\sigma)$ model performance is indeed comparable to the $(5\gamma 1\sigma)$ model performance ($\Delta\,\text{ELPD} = 0.74 \pm 0.9$; posterior mean $\pm$ std. err.). Moreover, the inferred population value of $\lambda$ in the $(5\gamma 1\lambda 1\sigma)$ model is consistent with 0: the posterior estimate is $\mu_\lambda = -0.17 \pm 0.19$ (mean $\pm$ std. err.), and the probability of direction is $p(\mu_\lambda < 0 \mid D) = 0.821$. The probability of direction is the proportion of the posterior distribution for a parameter that has the same sign as the median, and in simple regression scenarios it is related to a frequentist two-sided p-value ($p \simeq 2(1 - pd)$) [64].

## Supporting information

**S1 Fig. Rats are not sensitive to the temporal frequency of the unimodal, purely auditory stimuli.** Group average proportion of "High TF" choices (n = 10 rats) as a function of the TF of the unimodal auditory stimuli. Error bars are SEM.
(PNG)

**S2 Fig. The auditory stimuli and their intensity.** Each row refers to a distinct auditory stimulus used in the experiment: 1) the white noise burst with fixed maximal amplitude (top row; green); 2) the amplitude-modulated white noise bursts, with the 9 different temporal frequencies of the sinusoidal envelops (middle rows; gray); and 3) the absence of the auditory stimulus (bottom row; purple). **A**. Waveforms of the auditory stimuli used to drive the speakers that delivered the sounds to the rats (note that, for the sake of visualization, the sounds are not shown at their actual sampling frequency of 44.1 KHz but they have been downsampled to 500 Hz). **B**. Intensities of the auditory stimuli in dB as a function of time, as computed based on the waveforms shown in **A** and the measured intensity of 10 s long white noise bursts with different amplitudes (see the Materials and Methods for details). The panel referring to the TF of 0.25 Hz illustrates how the average sound intensity experienced by a rat in a given trial was

computed, based on the reaction time *RcT* of the animal. The latter was obtained by subtracting the estimated motor response time *MoReT* to the measured response time *RT*. **C**. Distributions and estimates of the average sound intensities experienced by each rat across all trials recorded for any given stimulus condition. Scatter plots show the estimated sound intensity experienced by the rats in each individual trial. Violin plots represent the full probability density distributions of these estimates while box plots show their median and interquartile range. The dashed line indicates the across-rat average intensity per stimulus condition.
(PNG)

**S3 Fig. Schematic of Bayesian hierarchical inference for our ideal observer model**. On the left, each row is labeled as representing the population parameters priors ($p(\eta)$), the subject-level parameters priors ($p(\theta \mid \eta)$) or the likelihood ($p(D \mid \theta)$). On the right, each bubble represents a parameter of the model. The name of the parameter and its distribution are reported. The rectangular plates indicate that multiple iid variables have been grouped. The group name and the corresponding number of variables are indicated on the bottom left of each plate. Each arrow represents the dependencies of the model. Gray coloring indicates that the variable is conditioned to observations.
(PNG)

**S4 Fig. Prior predictive simulation of the** $5\gamma$ $1\sigma$ **model**. In black the average proportion of answers "High" of the rats. Error bars denote the s.e.m. across all rats. Each colored line is the average predicted psychometric curve obtained for one of the five noise condition, generated by parameters sampled from its respective prior distributions. The $\gamma$ and $\sigma$ parameters are drawn from the population average parameters; the $\epsilon$ parameters, given the absence of a hierarchical structure, are averaged across all rats. The shaded areas represent one (dark area) and two (dim area) standard deviations from the mean. The five curves all look almost identical (up to sampling noise) as the priors for the noise conditions were the same.
(PNG)

**S5 Fig. Posterior predictive simulation of the** $5\gamma$ $1\sigma$ **model**. In black the average proportion of answers "High" of the rats. Error bars denote the s.e.m. across all rats. Each colored line is the average predicted psychometric curve obtained for one of the five noise condition, generated by the parameters sampled in the MCMC chain, following the respective posterior distributions (same as the solid lines displayed in Fig 5A). The $\gamma$ and $\sigma$ parameters are drawn from the population average parameters; the $\epsilon$ parameters, given the absence of a hierarchical structure, are averaged across all rats. The shaded areas represent one (dark area) and two (dim area) standard deviations from the mean of the posterior.
(PNG)

**S6 Fig. Posterior prediction of the linear relationship between sound intensity and perceptual gain factor in an ideal observer model**. The solid line represents the population-level gain parameter $\mu_\gamma$, modeled as a linear function of the average sound pressure level *I*. The slope of the line is $(-8.6 \pm 1.4) \times 10^{-3}$, the intercept $1.636 \pm 0.083$ (posterior mean $\pm$ st. dev.). The color gradient along the line maps the specific sound conditions: green for the fixed amplitude condition, grays for the V+AM conditions, and purple for the V-only condition. The dashed lines represent one standard deviation of $\mu_\gamma$. The shaded area around the regression line represents the posterior mean of the between-subject standard deviation, $\sigma_\gamma$. This area illustrates the estimated population-level variability. The symbols (triangle, square, cross) denote the relative average perceived sound pressure (x axis) and the posterior means (y axis) for individual rats. The inset shows the relative distance in Expected Log-Predictive

Density (ELPD) between the linear model and our best model ($5\gamma$, $1\sigma$), error bars denote the standard deviation of the difference.
(PNG)

**S1 Table.** Posterior summary statistics for the key population-level parameters of the model. The columns represent the posterior mean (**mean**), posterior standard deviation (**sd**), the 3% and 97% boundaries of the Highest Density Interval (**hdi_3%**, **hdi_97%**), as well as a number of standard diagnostics for MCMC-based Bayesian inference, namely the Monte Carlo Standard Error for the mean and standard deviation (**mcse_mean**, **mcse_sd**) [58], the bulk and tail Effective Sample Size (**ess_bulk**, **ess_tail**) [65], and the R-hat convergence diagnostic (**r_hat**) [65]. All diagnostics were computed with ArviZ version 0.15.1 [66]. Note that the $\mu_\epsilon$ values are not population parameters in the same hierarchical sense as the others; instead, they represent the simple average of the individual rat $\epsilon$ values estimated, as the $\epsilon$ parameters were modeled without a group-level distribution.
(CSV)

## Acknowledgments

We thank Mathew Diamond, Giuliano Iurilli, Maximiliano Jose Nigro, Matteo Marsili and Carlo Fantoni for valuable discussions and insightful feedback on the manuscript.

## Author contributions

**Conceptualization:** Mattia Zanzi, Francesco G. Rinaldi, Eugenio Piasini, Davide Zoccolan.

**Data curation:** Mattia Zanzi, Francesco G. Rinaldi.

**Formal analysis:** Mattia Zanzi, Francesco G. Rinaldi, Eugenio Piasini, Davide Zoccolan.

**Funding acquisition:** Eugenio Piasini, Davide Zoccolan.

**Investigation:** Mattia Zanzi, Silene Fornasaro.

**Methodology:** Mattia Zanzi, Francesco G. Rinaldi, Silene Fornasaro, Eugenio Piasini, Davide Zoccolan.

**Project administration:** Eugenio Piasini, Davide Zoccolan.

**Software:** Francesco G. Rinaldi.

**Supervision:** Eugenio Piasini, Davide Zoccolan.

**Visualization:** Mattia Zanzi, Francesco G. Rinaldi, Eugenio Piasini, Davide Zoccolan.

**Writing – original draft:** Mattia Zanzi, Francesco G. Rinaldi, Eugenio Piasini, Davide Zoccolan.

**Writing – review & editing:** Mattia Zanzi, Francesco G. Rinaldi, Silene Fornasaro, Eugenio Piasini, Davide Zoccolan.

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
