## [Decision Letter · Decision Letter 0]

20 Aug 2025

PCOMPBIOL-D-25-00990

Seeing what you hear: compression of rat visual perceptual space by task-irrelevant sounds

PLOS Computational Biology

Dear Dr. Zoccolan,

Thank you for submitting your manuscript to PLOS Computational Biology. After careful consideration, we feel that it has merit but does not fully meet PLOS Computational Biology's publication criteria as it currently stands. Therefore, we invite you to submit a revised version of the manuscript that addresses the points raised during the review process.

Please submit your revised manuscript within 60 days Oct 20 2025 11:59PM. If you will need more time than this to complete your revisions, please reply to this message or contact the journal office at ploscompbiol@plos.org. Please include the following items when submitting your revised manuscript:

We look forward to receiving your revised manuscript.

Kind regards,

Stefano Panzeri

Academic Editor

PLOS Computational Biology

Daniele Marinazzo

Section Editor

PLOS Computational Biology

**Journal Requirements:**

3) Some material included in your submission may be copyrighted. According to PLOSu2019s copyright policy, authors who use figures or other material (e.g., graphics, clipart, maps) from another author or copyright holder must demonstrate or obtain permission to publish this material under the Creative Commons Attribution 4.0 International (CC BY 4.0) License used by PLOS journals. Please closely review the details of PLOSu2019s copyright requirements here: PLOS Licenses and Copyright. If you need to request permissions from a copyright holder, you may use PLOS's Copyright Content Permission form.

Potential Copyright Issues:

i) Figures 1A, and 1B. Please confirm whether you drew the images / clip-art within the figure panels by hand. If you did not draw the images, please provide (a) a link to the source of the images or icons and their license / terms of use; or (b) written permission from the copyright holder to publish the images or icons under our CC BY 4.0 license. Alternatively, you may replace the images with open source alternatives. See these open source resources you may use to replace images / clip-art:

4) Thank you for stating "No competing interests to disclose." Please modify your Competing Interest statement on the submission form to the standard "The authors have declared that no competing interests exist."

**Reviewers' comments:**

Reviewer's Responses to Questions

**Comments to the Authors:**

**Please note that one of the reviews is uploaded as an attachment.**

Reviewer #1: uploaded as an attachment

Reviewer #2: In this study, the authors trained a group of rats in a visual temporal frequency classification task, in which visual stimuli were presented simultaneously with, but were task-irrelevant to, auditory stimuli. They developed a Bayesian ideal observer model, which captured the full spectrum of the rats’ perceptual choices. By integrating psychophysical experiments with computational modeling, the study identifies inhibition as a key mechanism underlying auditory–visual interactions.

Below are my comments:

(1) The authors employ a Bayesian ideal observer model, which is a commonly used approach. To improve clarity, it would be useful to specify whether any adaptations were made to the conventional model to suit the current experimental design.

(2) To validate the modeling approach, the authors should present goodness-of-fit measures that quantify the agreement between the model predictions and the observed behavioral data.

(3) The authors are encouraged to provide a clear summary of the critical model parameters, as this would significantly improve the transparency, reproducibility, and understanding of the modeling outcomes.

(4) The model's three underlying assumptions are outlined by the authors; however, further justification through citations from existing literature would enhance the theoretical foundation and clarify the rationale behind each assumption.

(5) In Figure 3, while rat 2 begins with a relatively high discrimination accuracy (~0.8), a noticeable decline occurs around training session 27, differing markedly from the trend observed in rat 1. The authors should provide a rationale for this discrepancy.

(6) The identification of inhibition as a key mediator of auditory–visual interactions represents a core conclusion of this study. To enhance conceptual clarity, the authors could consider providing a schematic diagram that illustrates the proposed inhibitory mechanism and its role in cross-modal processing.

Reviewer #3: It was a joy for me to read this excellent paper by Zanzi and Rinaldi et al. They report a carefully study on the effect of simultaneous task-irrelevant auditory stimulation on performance in a visual two-alternative forced choice discrimination task in rats. They find that auditory stimulation compresses the visual perceptual space in rats, and that increasing auditory intensity leads to increasing visual perceptual space compression. With impressive thoroughness, they construct a computational model that accounts for the observed data in a clear and straightforward manner. This study provides answers to several open questions about the nature of 'horizontal' (in terms of an inferential hierarchy) cortical connectivity between primary sensory cortical regions, and it clearly refutes some of the speculative explanations offered for observations from previous studies. In particular, Zanzi and Rinaldi et al. find that it is the total intensity of sensory stimulation prior to decision-making that determines the degree of compression of visual perceptual space, and more fundamentally that it is this compression which gives rise to the observed phenomena.

Since this study is extremely well-conceived and executed, and since the attendant modelling is entirely convincing, I believe it can be pubished more or less as submitted. Nonetheless, I have the following remarks:

- Looking at Figure 6B, wouldn't it have made sense to have a linear model (two parameters: intercept and slope) for the dependency of gamma on average sound pressure level instead of fitting a number of independent values for gamma and then embarking on a big model comparison exercise?

- While quantitative model comparisons of the kind reported can sometimes be helpful, it is generally more informative to choose a model on the basis of its performance in prior and posterior predictive simulations. It would therefore be interesting to see prior predictive simulations in addition the posterior ones in Figure 6A.

- Reinforcing the previous point, the priors of the chosen computational model should be justified by prior predictive simulation instead of an appeal to convention (line 857). Reference 54 (to which you appeal) would agree.

- Given the authors' impressive modelling capabilities, I was surprised to find some of the data analyzed by ANOVA, an unregularized out-of-the-box procedure.

**Have the authors made all data and (if applicable) computational code underlying the findings in their manuscript fully available?**

Reviewer #1: Yes

Reviewer #2: Yes

Reviewer #3: Yes

PLOS authors have the option to publish the peer review history of their article (what does this mean?). If published, this will include your full peer review and any attached files.

Reviewer #1: No

Reviewer #2: No

Reviewer #3: No

**Figure resubmission:**
---

## [Editor Report · Decision Letter 1]

10 Oct 2025

Dear Dr. Zoccolan,

We are pleased to inform you that your manuscript 'Seeing what you hear: compression of rat visual perceptual space by task-irrelevant sounds' has been provisionally accepted for publication in PLOS Computational Biology.

Best regards,

Stefano Panzeri

Academic Editor

PLOS Computational Biology

Daniele Marinazzo

Section Editor

PLOS Computational Biology

---

## [Editor Report · Acceptance letter]

PCOMPBIOL-D-25-00990R1

Seeing what you hear: compression of rat visual perceptual space by task-irrelevant sounds

Dear Dr Zoccolan,

I am pleased to inform you that your manuscript has been formally accepted for publication in PLOS Computational Biology. Your manuscript is now with our production department and you will be notified of the publication date in due course.

With kind regards,

Olena Szabo
